# Mineral reactivity determines root effects on soil organic carbon

Guopeng Liang [1,2], John Stark[1] & Bonnie Grace Waring [1,3] ✉

Modern conceptual models of soil organic carbon (SOC) cycling focus heavily on the microbe-mineral interactions that regulate C stabilization. However, the formation of 'stable' (i.e. slowly cycling) soil organic matter, which consists mainly of microbial residues associated with mineral surfaces, is inextricably linked to C loss through microbial respiration. Therefore, what is the net impact of microbial metabolism on the total quantity of C held in the soil? To address this question, we constructed artificial root-soil systems to identify controls on C cycling across the plant-microbe-mineral continuum, simultaneously quantifying the formation of mineral-associated C and SOC losses to respiration. Here we show that root exudates and minerals interacted to regulate these processes: while roots stimulated respiratory C losses and depleted mineral-associated C pools in low-activity clays, root exudates triggered formation of stable C in high-activity clays. Moreover, we observed a positive correlation between the formation of mineral-associated C and respiration. This suggests that the growth of slow-cycling C pools comes at the expense of C loss from the system.

What controls the size and turnover of the SOC pool? For decades, SOC stocks were thought to be regulated by plant traits, with more chemically complex inputs decomposing slowly and thereby constituting the C pools with the longest residence times. However, this paradigm has been overturned: we now recognize that soil C dynamics are largely governed by microbes and minerals[1,2]. Microbes are the primary conduit for SOC loss via their role in the breakdown of organic matter and its subsequent mineralization to $CO_2$. Yet most C with a long residence time consists of microbial products that are associated with clay mineral surfaces[3]. In other words, microbial physiology determines the way that new C inputs are partitioned between SOC formation and loss, impacting not only the total quantity of belowground C at any given point in time, but also its responses to future disturbance.

Our evolving understanding of the soil C cycle often focuses on organic matter 'stabilization,' i.e., the process by which SOC pools with long turnover times are formed[4–6]. However, C stabilization is not necessarily linked to the total *amount* of C in the soil at any given point in time. Particulate C pools, consisting largely of relatively decomposable plant detritus, turn over quickly and are highly vulnerable to perturbation (i.e. 'unstable') - yet they can accumulate indefinitely under the right conditions. In contrast, mineral-associated soil C is protected from microbial decomposition and therefore turns over more slowly (i.e., it is 'stable'), but there is an upper limit on this pool size, as further organo-mineral stabilization cannot occur once clay surfaces are saturated[7,8]. Moreover, C is always lost to respiration as it is transformed into the microbial precursors of slowly cycling SOC; thus, C is both stabilized and lost from the system simultaneously.

Increased soil C sequestration is a key component of efforts to stabilize the global climate[9], but lack of clarity around specific management objectives may impede this goal. Different strategies may be required to protect or enhance particulate vs. mineral-associated soil C pools. Moreover, although SOC stabilization and loss pathways are closely linked through microbial metabolism, there are multiple interacting drivers of SOC cycling along the mineral-microbe-plant

[1]Department of Biology, Utah State University, Logan, UT 84322, USA. [2]Present address: Department of Forest Resources, University of Minnesota, Saint Paul, MN 55108, USA. [3]Present address: Grantham Institute on Climate Change and the Environment and Georgina Mace Centre for the Living Planet, Imperial College London, London, UK. ✉e-mail: bonnie.waring@gmail.com

continuum that can influence microbial physiology. For example, chemical properties of the soil parent material dictate the capacity of soils to retain fresh C inputs, and shape microbial community structure as well[10]. There is increasing recognition that clay mineralogy is a better predictor of SOC dynamics than overall clay content: at global scale, highly reactive minerals, such as high surface-area phyllosilicate clays and oxyhydroxides, are associated with larger mineral-associated SOC pools[11,12]. By contrast, field-based and laboratory studies have found that microbial community structure has a larger impact on SOC dynamics than the chemistry of clay minerals[13,14], with fungi in particular playing a key role in the formation and turnover of particulate organic matter[15]. These divergent findings may reflect the different scales at which drivers of SOC were evaluated (global vs. regional), or perhaps the identity of the dominant SOC pool in each system being examined (i.e., particulate *vs.* mineral-associated C). For example, C cycle dynamics in soils with a high organic matter content might be more sensitive to microbial community dynamics than clay mineralogy, while the reverse should be true for C-poor soils.

Moving along the mineral-microbe-plant continuum, the literature reveals similar contradictions concerning the effects of plant input chemistry on SOC cycling and storage. The now well-accepted MEMS framework holds that the most bioavailable plant compounds (e.g. simple carbohydrates or amino acids) are more efficiently transformed into microbial biomass and subsequently into the mineral-associated C pool; thus, 'stable' SOC is dominantly derived from such high-quality inputs[1]. Multiple studies have demonstrated that the most energy- and nutrient-rich plant compounds are preferentially incorporated into the mineral-associated soil C pools[6,16,17]. However, this does not necessarily imply that such plant C inputs enhance *total* soil C stocks. A more growth-efficient microbial community can maintain a larger standing biomass at a given C input rate. A larger biomass, in turn, is associated with greater exoenzyme production, faster decomposition, and more SOC loss[18]. Therefore, when fresh C inputs accelerate C stabilization and loss simultaneously, there are uncertain consequences for the total C stock.

The fate of new C inputs to soil is determined not only by their chemical composition, but by their mechanism of delivery. Root exudates appear to play a particularly important role in SOC formation, for multiple reasons: exudates contain simple compounds that are preferentially assimilated by microbes[19], and enter the soil in close proximity to the minerals that may ultimately stabilise these microbial residues[20] (i.e., the 'microbial carbon pump'). At the same time, however, the high bioavailability of the organic matter in root exudates can also accelerate microbial decomposition of unprotected C via the priming effect[21], whereby fresh C inputs stimulate microbial growth, exoenzyme production, and increased mineralization of more complex, pre-existing organic matter[22]. Roots thus represent a 'double-edged sword'[23,24] which can increase or decrease the total amount of SOC depending upon the net impact of these two opposing mechanisms.

Few studies assess SOC stabilization and loss pathways simultaneously. The handful which do, often find divergent controls on these processes[24,25], but no study has explored all the critical drivers of C cycling along the plant-microbe-mineral continuum. Our goal was to evaluate the relative importance of mineralogical, microbial, and plant controls on two key processes that determine SOC pool size: mineral stabilization of new inputs, and respiratory soil C loss. To do so, we designed artificial root-soil systems to test three research questions. First, which is the principal driver of the quantity of C held in soil: microbes, which drive C loss through mineralization, or minerals, which protect C from microbial attack? We hypothesized (H1) that, in the mineral soils we studied, clay mineralogy would exert the dominant control, both directly (through organic matter sorption) and indirectly (through its influence on microbial community dynamics)[10]. We evaluated this hypothesis by creating artificial soils in which we could independently manipulate microbial assemblages (inoculating with whole soil communities or with bacteria only) and mineral composition (keeping clay content constant across all the soils, but varying mineral reactivity; Fig. 1). We then quantified microbial community shifts and soil C cycling over a three-month laboratory incubation. For

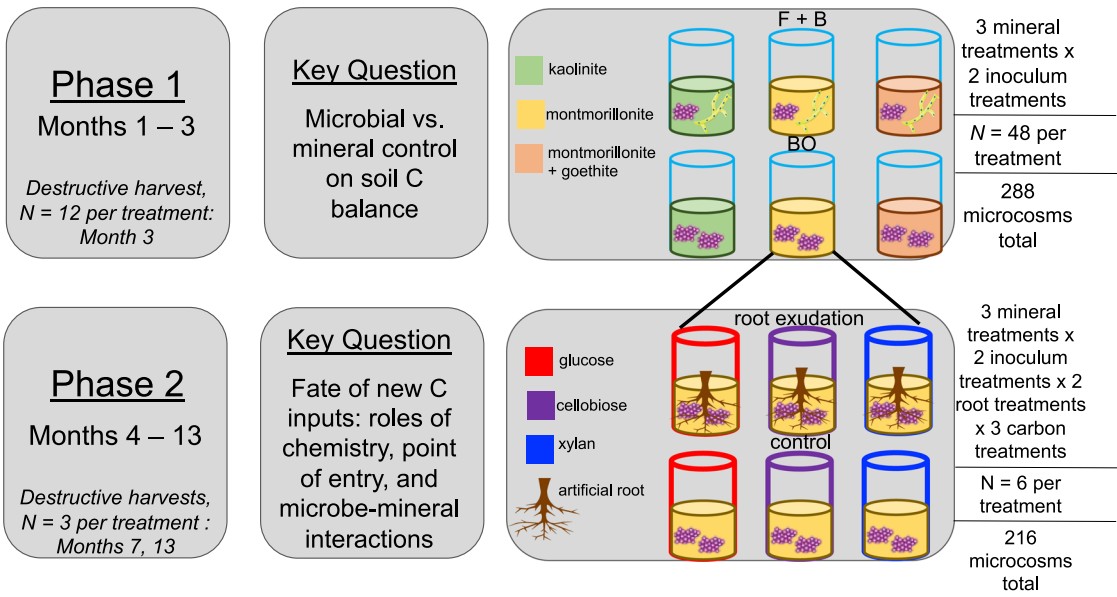

**Fig 1 | Experimental schematic.** Artificial root-soil systems were deployed to address questions about controls on soil C stabilization and loss. In the first experimental phase, we independently manipulated soil mineralogy and microbial communities, inoculating soils with either fungi and bacteria (FB) or bacteria only (BO) across three mineral reactivity treatments, in the presence or absence of artificial roots. There were 288 artificial soil microcosms in total, 72 of which were destructively harvested at the end of the first phase to quantify soil C pools. In the second phase, experimental units from each unique mineral x microbe treatment combination were further modified through a fully factorial manipulation of aboveground C input type (glucose, cellobiose, xylan) and enhanced root exudation through mimetic root systems (exudates vs. water-only control). This created 36 unique treatment combinations, each replicated 6 times, with 3 microcosms in each unique treatment combination destructively harvested at month 7 or 13. We measured soil $CO_2$ loss throughout the 13-month incubation. Note that 108 'real soil' microcosms were also incubated and subject to the same inoculum, root, and carbon amendment treatments – this control experiment is not visualized here.

the second question we asked, how do the chemistry and point-of-entry of different C inputs affect their interactions with microbes and minerals and thus, the subsequent fate of those inputs in the soil? We were particularly interested in whether the effect of root C inputs on overall SOC stocks is dependent upon the chemical complexity of non-root C inputs. To address this question, we amended surface soils with organic matter varying in chemical composition, either with or without additional inputs through artificial root exudation, and followed the microcosms for a further 10 months (Fig. 1). Therefore, in its second phase, the experiment consisted of a fully factorial manipulation of clay mineralogy, initial microbial inoculum, the chemistry of surface C amendments (ranging from the monomer glucose to the complex polymer xylan), and root exudation (present or absent). We hypothesized that, in line with current conceptual frameworks[1], the most bioavailable C inputs would be preferentially incorporated into microbial products. This should result in a larger biomass and mineral-associated C pool, especially in the presence of reactive minerals, but also larger respiratory C losses (H2a). We also expected simple C delivered via root exudates to enhance the decomposition and subsequent mineralization of more chemically complex inputs (H2b). For our third and final question, we asked, what are the relationships among C stabilization, C mineralization, and total SOC pools across all experimental treatments? Here, we define C 'stabilization' as the formation of mineral-associated C. Although under some circumstances minerals accelerate soil C loss[26], on average mineral-associated organic matter turns over up to 1000 times more slowly than particulate organic matter[12]. We hypothesized (H3) that across all experimental treatments, the formation of mineral-associated C would be negatively correlated with C loss through respiration. This is because organic matter captured in the slower-cycling mineral-associated C pool is protected from microbial attack, reducing the total quantity of C available for mineralization.

Here we show, using artificial root-soil systems, that the formation of mineral-associated SOC is controlled by the interaction between mineral surfaces, microbes, and root exudates. In contrast with current conceptual models, the chemical complexity of individual C compounds did not impact mineral-associated C formation or the composition of microbial communities. Across all treatments, soil C stabilization was *positively* correlated with C loss, demonstrating how subtle changes in total SOC stocks belie large changes in the underlying fluxes.

## Results and Discussion
### Mineral versus microbial controls on soil carbon loss
We constructed artificial soils with equivalent organic matter and clay contents, but with a broad spectrum of mineral reactivities (low reactivity: kaolinite; medium: montmorillonite; high: montmorillonite + goethite). By manipulating the composition of the initial inoculum (fungi + bacteria [FB] *vs.* bacteria-only [BO]), we also generated differences in the microbial communities inhabiting the artificial soils. To explore root effects on soil properties, half the microcosms in each clay activity/inoculum treatment group received exudates (3.72 μg C g$^{-1}$ d$^{-1}$), delivered through an artificial root system (Fig. S1) which mimicked realistic spatial patterns of root C release. The remaining microcosms had artificial roots installed, but received only sterile water. After three months of incubation, artificial soils came to resemble real (non-artificial) soils incubated under identical conditions, exhibiting similar levels of microbial biomass, enzyme activity, and fungal (but not bacterial) species richness (Fig S2a–e).

Contrary to our hypothesis (H1), during the first three months mineral reactivity did not influence soil respiration, but the total C pool in each microcosm (calculated via a mass-balance approach; see Methods) was 4% larger in the FB vs. BO treatment (Table S1a). This is because microcosms in the BO treatment exhibited 20% greater rates of respiration (Table S1, Fig S3). However, inoculum-related

differences in respiration rate diminished through the course of the first experimental phase (Fig S3a, b). While the inoculum treatment had a significant (albeit small) impact on respiration, its effects on the taxonomic composition of microbial assemblages were profound (Table S2, Fig S4). Microcosms in the BO inoculum treatment were characterized by substantially less diverse bacterial and fungal communities, with species richness reduced by four-fold and two-fold, respectively (Table S1). Nearly all fungal taxa identified in the BO treatment belonged to a single class, the Eurotiomycetes (Fig S5). Microbial communities in the BO treatment also had 11% higher C use efficiency (Table S1b).

Although soil mineralogy did not influence the size of the SOC pool, it did have a pronounced impact on the composition of microbial communities, as has been observed elsewhere[10,27], with assemblages on kaolinite clays compositionally and functionally distinct from those in the montmorillonite-dominated soils (Fig S4). Kaolinite soils were characterized by much greater bacterial taxonomic diversity, a lower microbial C use efficiency, and a smaller standing microbial biomass (Table S1, Fig S2a–e). Meanwhile, low levels of root exudation supplied during months 0–3 had only minor effects on soil respiration and microbial communities (Tables S1, S2). Thus, in the early stages of artificial soil development, microbial community structure had the strongest impact on C losses through respiration and potentially, C stabilization.

### Fate of new C inputs: role of chemistry, point-of-entry, and microbe-mineral interactions
In the second phase of the experiment, which began 3 months after the microcosms were established, we began adding different forms of C to the soil surface and augmented rates of root exudation, thereby manipulating both the chemical quality and point of entry of the inputs. All microcosms received additional C at a rate of 27.5 μg C g$^{-1}$ d$^{-1}$, but the chemical complexity of this C varied: a third of the experimental units received glucose, a third received cellobiose, and a third received xylan. These inputs were added to the surface of the soil to simulate fluxes of C from aboveground litter into the soil profile. The quantity of C delivered through the artificial root systems to microcosms in the root exudate treatment was also increased 10-fold, to 37.2 μg C g$^{-1}$ d$^{-1}$. During this second experimental phase, we observed a 70% increase in respiration rates and the formation of a mineral-associated C (MAOC) pool, reflecting the stabilization of, on average, 20.9% of the total C available to each microcosm. Across treatments, MAOC accounted for 47.7 ± 1.0% of total SOC, a proportion quite comparable to the global mean for terrestrial soils (where 65% of SOC is mineral-bound)[28]. The microbial biomass also accounted for a comparatively large fraction (30.8 ± 0.7%) of the total SOC pool. This likely reflects the high rate of C inputs, as the figure was similar (16.3 ± 0.6%) even in the real-soil controls.

In the second experimental phase, soil inoculum composition continued to influence soil C cycling (Table S3). Although respiration rates did not differ between inoculum treatments in the second experimental phase, MAOC pools were larger in microcosms which were inoculated with FB *vs.* BO (Fig S6), especially in the cellulose treatment. This corroborates new evidence that fungi play a critical role in stable SOC formation[29]. Moreover, both $CO_2$ fluxes and MAOC were greatest in soils containing goethite (Fig S6). This suggests that metal oxides accelerated C stabilization and loss simultaneously, promoting the formation of organo-mineral bonds, but perhaps also coupling iron reduction with C mineralization in anoxic soil microsites[26].

We expected to find an overall trend towards lower respiration, microbial biomass and MAOC in the xylan treatment, reflecting slower decomposition and less efficient conversion of plant biomolecules into microbial tissue (H2a). However, these predictions were only partially supported. While respiration rates were 41% lower in microcosms amended with xylan vs. cellobiose or glucose (Fig S6), the microbial

biomass was actually slightly *larger* in the xylan treatment than in the glucose treatment (Fig S6), and the chemical complexity of above-ground C inputs had no impact on MAOC formation (Table S3, Fig S6). Although fungal communities tended to be marginally more species-rich in the xylan treatment (Fig S7), the overall impacts of C input chemistry on bacterial and fungal communities were extremely weak (Tables S3, S4). These data do not support newer conceptual frameworks which posit linkages among the chemistry of C inputs, the growth efficiency of the microbial biomass, and C stabilization[1]. This is surprising, as similar experiments have shown profound impacts of C input type on microbes and soil organic matter formation[13,30]. In contrast with those studies, we did not observe substantial shifts in the taxonomic composition of bacterial or fungal communities across the different C input treatments. Changes in microbial physiology might buffer MAOC formation against variation in input chemistry[13]. For example, although C mineralization rates were lower in the xylan treatment, these microbial communities maintained a relatively large biomass, suggesting greater growth efficiency. This, in turn minimized cross-treatment differences in the quantity of microbial tissue available for mineral stabilization. It is also possible that amending soils with individual compounds (rather than whole plant litters) obscured ecological dynamics observed in real ecosystems, e.g. microbial community specialization on particular plant types.

We anticipated strong interactions between the root treatment and the composition of non-root C inputs, with bioavailable exudates accelerating decomposition of the complex polymer xylan, but not glucose or cellobiose - i.e., the priming effect (H2b). This hypothesis was also unsupported, as interactions between C input chemistry and root exudation treatments were generally insignificant for biogeochemical variables (Table S3). However, MAOC pools in microcosms amended with glucose were 25% lower in the presence of root exudates. Diversity and structure of bacterial communities responded to a root-carbon interaction as well (Tables S3, S4), suggesting this MAOC response was microbially mediated, perhaps reflecting a shift towards a fast-growing copiotrophic community.

Nearly all biogeochemical parameters measured were strongly affected by an interaction between root exudates and soil mineralogy (Table S3, Fig. 2). Root exudates stimulated respiration in all microcosms, increasing $CO_2$ flux by 29% on average, but this effect diminished as mineral reactivity increased. Root exudates also enhanced the microbial biomass C pool by approximately 13%, but this effect *increased* as mineral reactivity increased. Finally, soils in the root exudate treatment had 42–26% less MAOC in kaolinite and montmorillonite soils (in comparison with paired no-exudate controls). Yet in soils containing goethite, MAOC pools were 10% larger in the root exudate treatment (Fig. 2). These patterns were modified to some extent by the composition of the microbial community and by non-root C inputs: for example, root exudates increased C sorbed to montmorillonite+goethite in the BO but not the FB treatment, and when additional C was added as cellobiose or xylan but not glucose. These C cycling patterns were accompanied by reorganization of bacterial and fungal communities in response to root exudation, with different trajectories depending upon mineralogy (Table S4, Fig. 3). In general, both bacterial and fungal communities were significantly more diverse in the presence of root exudates (Table S3, Fig S7), and tended to have a higher abundance of the Actinobacteria, Firmicutes, and Bacteriodetes that characterized real soils (Fig S8).

Taken together, our data illustrate how soil mineralogy mediates the effect of root exudates on C stabilization and loss mechanisms, shaping both the size and chemical composition of the SOC pool (Fig. 2). Soils containing goethite exhibited larger MAOC pools than those consisting of clay minerals only, demonstrating the superior C stabilization capacity of metal oxides in comparison with phyllosilicate clays across a broad range of soil conditions[31–33]. Moreover, only in the presence of goethite did exudates enhance, rather than

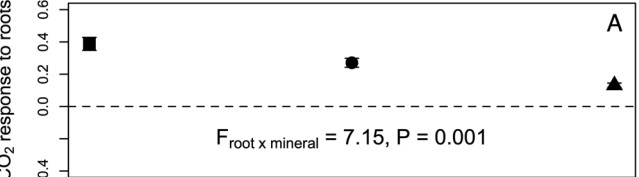

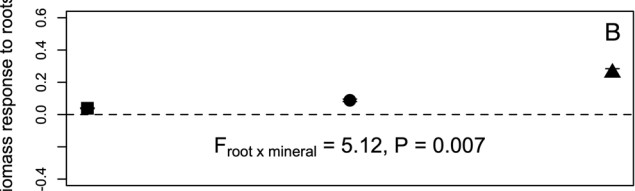

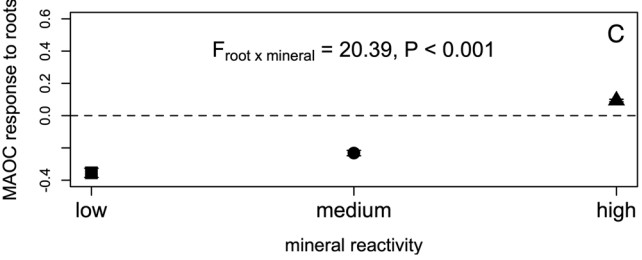

**Fig. 2 | Root-mineral interactions affect soil carbon pools.** Mean log response ratios visualising the significant interaction between mineral reactivity and root exudate treatments for cumulative $CO_2$ fluxes (**A**), microbial biomass pools (**B**), and mineral-associated organic C [MAOC] pools (**C**). Data are shown for the second experimental phase, with elevated rates of root exudation. Log response ratios are calculated as the log of the ratio in the exudate treatment *vs.* no-root control; thus, values greater than 0 indicate an increase in the corresponding parameter under root exudation. $N = 36$ in each treatment. Error bars represent standard errors, which incorporate variance in both root exudate and control treatments.

diminish, MAOC formation. These patterns may shed light on the mineral traits that govern the stabilization of root exudates. Kaolinite and montmorillonite had very similar impacts on MAOC formation, despite their very different surface areas and cation exchange capacities; this suggests that neither CEC nor clay surface area strongly influenced the fate of C inputs in our artificial soils. However, both phyllosilicate minerals differ from goethite in a critical respect: in contrast with 1:1 phyllosilicate minerals like kaolinite and 2:1 phyllosilicate minerals like montmorillonite, which are negatively charged under normal soil pH conditions[34], metal oxides like goethite exhibit variable charge and bear highly reactive hydroxyl groups[35]. Bonds formed through ligand exchange between the hydroxyl groups of organic matter and metal oxides are among the most durable[36], and explain the particular affinity of hydroxyl/carboxyl-rich compounds for goethite[37,38]. Although the simulated root exudate solution we added did not contain these compounds, it did provide simple sugars that fuel microbial growth and production of carboxyl/hydroxyl-rich metabolites, e.g. oxalic acid[39] or glutamate. By contrast, the compounds most likely to be stabilized on negatively charged phyllosilicate clays, primarily via ionic bonds, would include basic amino acids (which were included in the simulated root exudate), or microbial residues stabilized via cation bridging. Both of these bonding mechanisms are weaker than the ligand exchange observed on goethite. Therefore, we suggest that root-induced acceleration of microbial growth promoted stabilization of biomass residues on metal oxide surfaces, but disrupted weaker organo-mineral bonds in phyllosilicate clays.

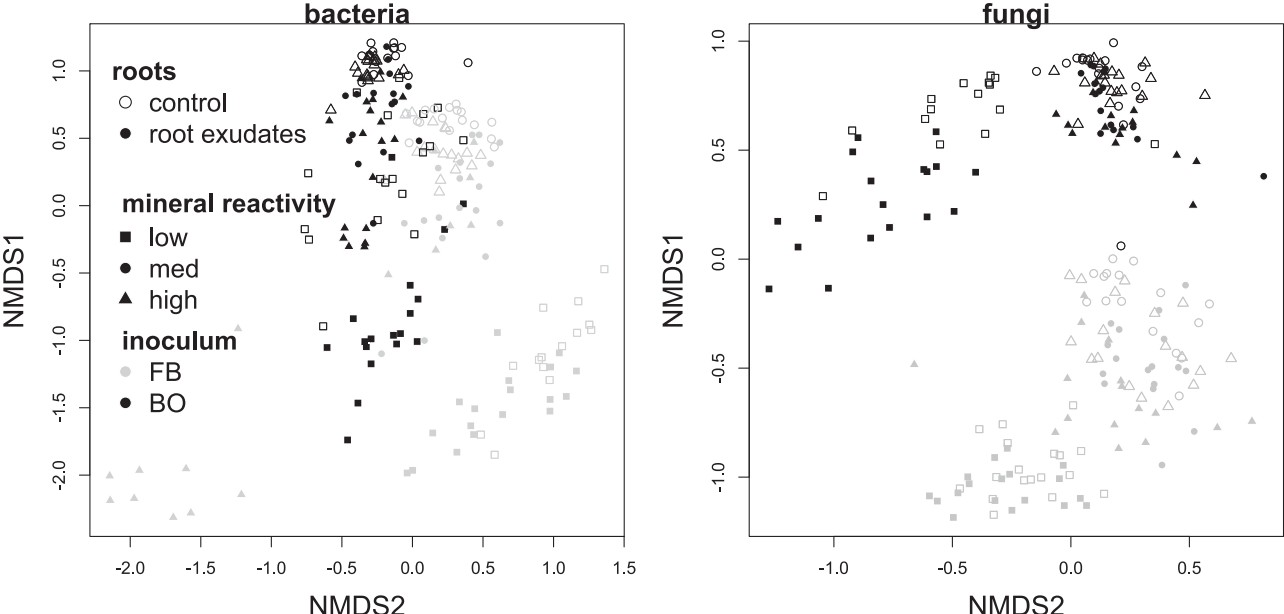

**Fig. 3 | Treatment effects on microbial community structure.** Non-metric multidimensional scaling analysis conducted on a Bray-Curtis dissimilarity matrix of both bacterial and fungal communities in the second phase of the experiment, demonstrating the strong impact of the microbial inoculum treatment (black vs. gray symbols), mineral reactivity (indicated by symbol shape), and root exudation (open/closed symbols) on bacterial and fungal assemblages. Data for all 216 microcosms in the second phase of the experiment are shown. Treatments and treatment interactions which explained < 5% of compositional variance are not visualized here.

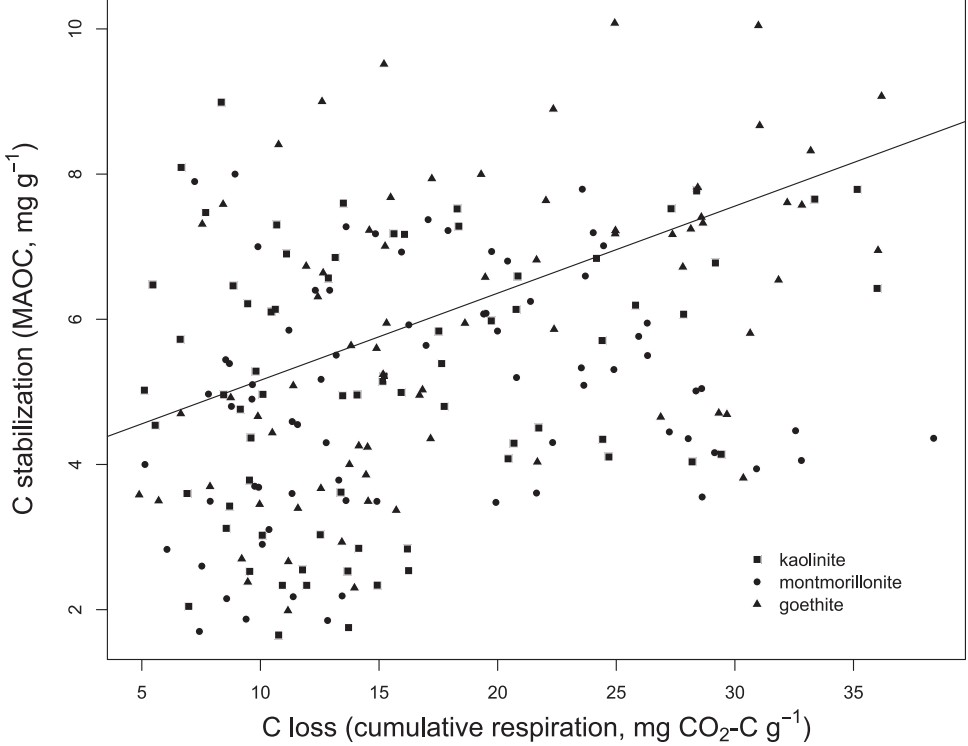

**Fig. 4 | Relationship between soil carbon stabilisation and loss.** Linear regression showing the relationship ($R^2 = 0.131$, $P = 0.029$) between an index of C stabilization (pools of mineral-associated organic C, or MAOC) and loss (cumulative respiration) in 216 microcosms included in the second phase of the experiment. The quantity of C input to each microcosm was included as a covariate.

## Relationship of soil C stabilization and soil C loss

When viewed holistically, data from our experiment suggest that root exudates, microbes, and minerals interact to regulate both soil C loss via respiration and stabilization via MAOC formation. These processes are often assumed – implicitly or explicitly – to be negatively correlated, such that an increase in soil C loss by respiration will reduce the availability of organic matter for stabilization (or, conversely, that C capture in MAOC reduces the amount of substrate available for mineralization) (H3). This assumption can be challenging to validate empirically, because it requires the fate of all organic matter entering

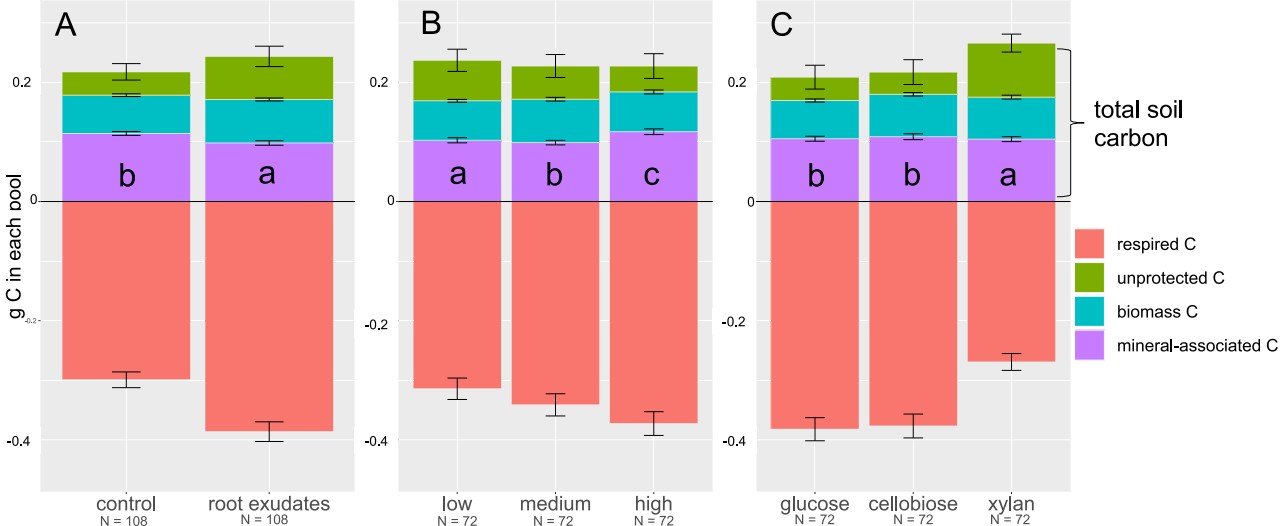

**Fig. 5 | Fate of carbon in different experimental treatments.** Distribution of mean C pools in artificial soils (in g of C per microcosm) as a function of experimental treatment: (**A**) Root exudates, (**B**) mineral reactivity, and (**C**) aboveground C amendment. All data shown are for artificial soils in the second phase of the experiment. All pools were directly and independently quantified with the exception of the unprotected (non-mineral associated) particulate organic carbon pool, which was determined via a mass balance approach. Error bars reflect standard errors; for the unprotected C pool, these errors are propagated across all directly measured pools. Letters on the graphs indicate significant differences among treatment levels in Tukey's post-hoc tests, conducted following analysis of variance on total soil carbon.

and leaving the ecosystem to be known. In our artificial soil systems, respiration represents the only pathway of soil C loss, and C stabilization could be quantified in a straightforward way as the accumulation of organic matter on initially unoccupied mineral surfaces. We explored the relationship between these two processes using a linear regression, controlling for the total quantity of C inputs to each microcosm in each time interval. We found that C losses (cumulative respiration) were *positively* correlated with C stabilization (MAOC) ($R^2 = 0.131$, $P = 0.029$; Fig. 4). We suggest that this relationship emerges from fundamental constraints on microbial metabolism: synthesis of the tissues and exudates which are ultimately stabilized is necessarily accompanied by microbial respiration[2]. Therefore, C stabilization through the "microbial carbon pump" always occurs at the expense of soil C loss.

What are the implications of these findings for C sequestration potential of artificial soils? Using a mass-balance approach, we found that more soil organic matter accumulated in the presence of root exudates, in low activity clays, and under amendment with xylan (Table S5, Fig. 5). The relationship between total soil C stocks and respiratory C losses was inconsistent across these treatments, potentially reflecting differences in the dominant source of the C that was respired (pre-existing organic matter vs. fresh C inputs via root exudation or surface amendment). Although soils in the root exudate treatment respired more in total, they also received greater C inputs than soils in the no-root control treatment. Total C inputs exceeded C losses, leading to net growth of the total SOC pool (Fig. 5A). By contrast, soils with low-activity clays and under xylan addition respired less than their counterparts, which allowed a larger proportion of C inputs to be retained (Fig. 5B, C). These data show how overall rates of soil C cycling can be decoupled from changes in the size of the soil organic matter pool. Moreover, the treatments with the largest total C stocks also exhibited the smallest MAOC pools (Fig. 5). This clearly illustrates that the size of the SOC pool is not dictated by the proportion of this carbon that is 'stable.'

### Conclusions and ecosystem implications

Our artificial root-soil systems were created as a tool to explore fundamental constraints on SOC cycling, and do not capture the full complexity of processes occurring in real soils. However, our experiment revealed the mechanisms underlying many patterns that are observed at much larger scales: the acceleration of C cycling[21] and changes in MAOC[40,41] induced by root exudation; as well as the importance of minerals in shaping microbial community assembly[10]. Our finding that C loss and stabilization are positively correlated with each other, but uncorrelated with total C stocks, illustrates a key point that is also applicable to real ecosystems. Having a greater proportion of 'stable' vs. rapidly-cycling organic matter will slow the transit time of C atoms through the system, but it does not necessarily mean that there will be *more* carbon present in the system at a particular point in time. This distinction is critical in light of efforts to enhance the soil C stock as a mechanism of climate change mitigation[9]. On the one hand, management interventions that accelerate C stabilization will reduce the vulnerability of SOC to global change[42]. However, these same interventions may not enhance total SOC stocks[7], and thereby the amount of C removed from the atmosphere. Protecting large, vulnerable, 'unstable' pools of SOC is also critical to limit atmospheric $CO_2$ increase[43,44].

## Methods

### Experimental design

We developed artificial root-soil systems to conduct a 13-month microcosm incubation experiment with a fully factorial manipulation of clay mineralogy, soil microbial community composition, presence or absence of root exudates, and aboveground C input quality ($N = 288$ microcosms in total; Fig. S9). In order to benchmark the performance of the artificial soils, a real soil with a similar organic C and clay content was also incubated under the same microbial, root, and C input treatments ($N = 108$ in total). To create the clay mineralogy treatments, we used organic-matter-free quartz sand, pure clay minerals, and ground corn leaves (sterilized via autoclaving) to construct sterile artificial soils with equivalent clay contents ($0.1\ kg\ clay\ kg^{-1}$ soil) and C concentrations ($0.017\ kg\ C\ kg^{-1}$ soil), but different levels of mineral reactivity (Table S6), as defined by surface area and charge. Inorganic nutrients were added to each microcosm to bring initial soil carbon:nitrogen and nitrogen:phosphorus ratios to 25:1 and 8:1, respectively, after accounting for nutrient content of the ground corn tissue.

Assuming a microbial CUE of approximately 0.5, we intended this to result in an artificial soil organic matter C:N of 12.5:1, approaching the global average for mineral soils[45]. The 'real-soil' control we used was a coarse-silty Xeric Haplocalcid, collected from a sagebrush steppe in northern Utah, with a soil C concentration of 0.0173 kg kg$^{-1}$ soil and clay content of 0.121 kg kg$^{-1}$ soil. This soil was sterilized via double-autoclaving prior to microcosm construction.

Both artificial and real soils were sterilized at the start of the incubation, and microcosms were randomly assigned to one of two microbial inoculum treatments: one containing both fungal and bacterial propagules ('fungi + bacteria'), and the other excluding most fungal hyphae and spores ('bacteria only'). Both inocula were generated from a slurry of 5 g of compost (EcoScraps, Marysville, OH) blended in 1 L sterile water. The bacteria-only inoculum was additionally filtered through a 2 μm filter and treated with the antifungal captan (0.05 mg mL$^{-1}$).

In order to mimic realistic rates of root exudation and root-mineral interaction, we modified the methods of[46,47] to create a gravity-fed artificial root system (Fig.S1a, b). Each artificial root system consisted of ten, 10 cm long polysulfone hollow fiber dialysis membranes (HFMs) (Fresenius Medical Care North America, Ogden, UT) connected to a 5 mL plastic reservoir containing root exudate solutions. The HFMs were characterized by a lumen diameter of < 200 μm and wall thickness of < 50 μm, allowing the exchange of biomolecules < 60 kDa across either side of the membrane. The artificial root systems were positioned such that the bottom of the reservoir, from which the HFMs emerged, was flush with the soil surface. All artificial soil microcosms had artificial roots, but only two-thirds of the real-soil microcosms did; the remaining real-soil microcosms were each planted with five sterile *Lolium perenne* seedlings. This helped us evaluate whether real and simulated root exudations affect soils in similar ways (e.g. see Fig S1c).

The experiment consisted of two phases. In Phase 1 (months 1–3), C inputs were restricted to the exudate solution delivered to the microcosms as part of the 'root exudation' treatment (i.e., half the artificial soil microcosms and 1/3 of the real soil microcosms). The 'root control' microcosms received only sterile water through the artificial root systems. The exudate solution consisted of glucose, fructose, sucrose, and tryptic soy broth[46], delivered at a rate of 3.7 μg C g$^{-1}$ soil d$^{-1}$ to match observed rates of exudation quantified in situ[48,49]. At the end of Phase 1, six replicates of each treatment combination were destructively harvested for analysis of microbial biomass, extracellular enzyme activities, and microbial C use efficiency. In Phase 2 (months 4–13), we increased exudation rates in the 'root exudate' treatment by an order of magnitude to 37 μg C g$^{-1}$ d$^{-1}$. Additionally, microcosms were randomly assigned to a non-root C input treatment, applied directly to the soil surface, to establish variation in the chemical complexity of aboveground inputs: glucose (monosaccharide), cellobiose (disaccharide), or xylan (polysaccharide). Each compound was mixed in aqueous solution with NH$_4$Cl to adjust the carbon:nitrogen ratio to 25:1, and supplied at a rate of 27 μg C g$^{-1}$ soil d$^{-1}$ (which is at the high end of the range of soil C inputs in very productive forests[50]). Microcosms were amended with the appropriate C solution every two weeks, the same time interval over which artificial root reservoirs were emptied and replaced with fresh exudate solution. In the second phase, destructive harvests (N = 3 per unique treatment combination for artificial soils, and N = 2 for real soils) were conducted at 7 and 13 months.

## Quantifying C pools and fluxes

CO$_2$ fluxes were measured at two-week intervals throughout the experiment. Microcosms were tightly capped for 24 h prior to headspace sampling with a gastight needle, and the headspace samples were analyzed using gas chromatography (GC-2016 Greenhouse Gas Analyzer, Shimadzu, Kyoto, Japan). Additionally, at the end of Phase 1, we measured C use efficiency (CUE) of the microbial community to

determine if clay, root exudate, or inoculum treatments affected this parameter. The CUE was measured by adding 0.116 mg C g$^{-1}$ soil of a glucose solution with an enrichment of 26 atom % $^{13}$C. Headspace $^{13}$C-CO$_2$ concentrations were measured during the following 24 h using a G2131-i Isotope and Gas Concentration Analyzer (Picaro Inc., Santa Clara, CA, USA). To estimate $^{13}$C assimilation by microbes, we also measured 'residual' $^{13}$C in chloroform-fumigated and unfumigated solid soils following extraction of dissolved C with 0.5 M K$_2$SO$_4$. This method assumes that there is no abiotic sorption of glucose and that all of the non-extractable $^{13}$C resulted from microbial assimilation[51]. We calculated soil microbial CUE as $dB_C / (dB_C + \sum CO2 - C)$, where $dB_C$ is the amount of glucose-C incorporated into microbial biomass, and $\sum CO2 - C$ is the cumulative respiration of glucose C over 24 h. Finally, at months 7 and 13, we measured microbial biomass and mineral-associated organic C pools, extracellular enzyme activities, and inorganic N concentrations. Microbial biomass was measured via chloroform fumigation and direct extraction with 0.5 M K$_2$SO$_4$[52] while MAOC was quantified as the C content in the soil 'heavy fraction' following soil density fractionation with sodium polytungstate (1.6 g cc$^{-1}$)[53]. On average, we recovered 100.4 ± 0.2 % of the mass of soil subject to density fractionation - that is, the mass of the isolated heavy and light fractions summed to the mass of the whole soil sample. This indicates near-perfect preservation of the sample (no C loss) and very minor contamination of the sample with sodium polytungstate. C concentrations and isotopic signatures in heavy fraction samples were quantified on a Costech 4010 Elemental Analyzer coupled to a Thermo Scientific Delta V IRMS. Because light fraction masses were so low, we did not quantify C content of these samples directly. Rather, the quantity of 'unprotected C' in each microcosm (in particulate and dissolved organic matter) was calculated via a mass balance approach. We determined 'total available C' in each microcosm as the sum of C contained in the ground corn leaves, root exudates and surface amendments of glucose, cellobiose, or xylan. We then calculated 'recovered C' – the sum of cumulative C lost to respiration, microbial biomass C, and heavy fraction C (MAOC) – quantities determined empirically for each microcosm. The difference between total available C and recovered C was termed 'unprotected C,' and is assumed to reflect particulate C that was not incorporated into the microbial biomass or stabilized on mineral surfaces at the time of measurement. Furthermore, we refer to the difference between 'total available C' and cumulative respiration as the 'total C pool' in each microcosm – this invokes the assumption that no C losses occurred other than through respiration (which is appropriate as the microcosms were closed systems). Activities of the extracellular enzymes including β-xylosidase, acid phosphatase, β-glucosidase, cellobiohydrolase, leucine aminopeptidase, and β-N-acetylglucosaminidase were determined following[54].

## Microbial community analyses

We used a DNeasy PowerSoil Kit (Qiagen, Hilden, Germany) to extract DNA from all microcosms; the DNA extracts were sequenced on the Illumina MiSeq platform at the Utah State University Center for Integrated Biosystems. In order to amplify bacterial and fungal marker genes, primers 515F−806 R and ITS1f-ITS2 were used, respectively. We used the QIIME 2 (version 2022.2)[55,56] to analyze the sequencing data. The DADA2 algorithm[57] was used to denoise sequences and produce putatively error-free amplicon sequence variants (ASVs). We used the UNITE[58] and Greengenes training datasets[59] to assign fungal and bacterial taxonomy.

## Statistical analyses

Phase 1 and 2 of the experiment were analyzed separately, as distinct experimental manipulations were performed in each. Three-way ANOVAs were performed to determine effects of artificial root exudates, soil microbial community composition, and soil mineralogy on response variables in Phase 1. ANOVAs on Phase 2 data additionally

incorporated effects of non-root C input and date of destructive microcosm harvest (7 months or 13 months). Statistically significant outliers (identified with the 'outlierTest' function in R package *car*) were removed, and data transformed as required to meet conditions of normality and heteroskedasticity. We analyzed $CO_2$ data using mixed models with microcosm identity as the random effect (*lme4*[60]); we also calculated cumulative $CO_2$ respired in each microcosm using the 'area under curve' function in the R package *flux*, which integrates under the curve of $CO_2$ flux vs. time following the trapezoid rule[61]. Finally, we visualized soil microbial community data using non-metric multi-dimensional scaling (NMDS) which was performed on Bray-Curtis dissimilarity matrices of ASV abundances; permutational ANOVAs were used to assess the significance of treatment effects on microbial community composition (i.e., presence and relative abundance of taxa). All analyses were conducted in R version 4.1.1.

### Reporting summary
Further information on research design is available in the Nature Portfolio Reporting Summary linked to this article.

## Data availability
The biogeochemical data generated in this study have been deposited at FigShare, which can be accessed at the following links: https://doi.org/10.6084/m9.figshare.21333048.v1[62], https://doi.org/10.6084/m9.figshare.23807352.v1[63], and https://doi.org/10.6084/m9.figshare.23807355.v1[64]. Sequence data are available at the NCBI Sequence Read Archive, http://www.ncbi.nlm.nih.gov/bioproject/1001150.

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

## Acknowledgements

We thank Preston Christensen, Karen Foley, Jalynn Jones, and Camilla Moses for laboratory assistance. This work was funded by the U.S. Department of Energy DE-SC0020108.

## Author contributions

B.G.W. conceptualized and designed the study, and G.L. collected the data. B.G.W., G.L., and J.S. analysed the data. B.G.W. wrote the manuscript with substantial contributions from all co-authors.

## Competing interests

The authors declare no competing interests.
