## [Peer Review File · Nature Communications]

Mineral reactivity determines root effects on the stabilization and mineralization of soil organic carbonReviewer #1 (Remarks to the Author):

MANUSCRIPT BRIEF

General Problem

26: What controls the size and persistence of the SOC pool?

Specific Research Question

15: What is the net impact of microbial metabolism on the total quantity of C held in the soil?

Scientific Unknown

59: We lack a quantitative framework to predict how a change in root production will impact SOC stocks

70: when labile C inputs accelerate C stabilization and loss simultaneously, there are uncertain consequences for the total C stock

Objectives

The scientific objectives pursued with this study were not identified

Hypotheses

The authors do not identify the hypotheses they were presumably testing. They must have assumed that variations in root influence, microbial community composition, mineralogy and substrate chemistry would lead to differential outcomes. It is not clear why they choose not to share their assumptions = hypotheses.

If the study was meant to be purely exploratory than this should be stated and rationalized.

Conceptual Approach

The only statement giving some level of strategic reasoning is found in the abstract:

16: we constructed artificial root-soil systems to identify controls on C cycling across the plant-microbe-mineral continuum, simultaneously quantifying the formation of persistent mineral-associated C and C losses to respiration. There is no rationale / justification / deduction explaining the choice of the three experimental treatments (root presence; mineralogy and microbial community composition)

The latter two factors are labeled as "soil forming" (94). The authors are invited to review contemporary reviews of soil forming factors to learn that both, mineralogy and microbial community composition are the result of pedogenesis and not its cause.

A description of the rather complex experimental setup follows 83-91, but rationalization does not go beyond 87 "to explore the effect on....."

Results

Figure 1: illustrates influence of root exudates on

- cumulative C fluxes (decreasing with increasing clay activity)
- microbial biomass (increasing with increasing clay activity) and
- Mineral associated C pools (increasing with increasing clay activity)

Figure 2: Has two panels, one for bacteria and one for fungi. The caption is incomplete/inadequate in that it identifies the statistic procedure performed, but omits any hint at what point the panels are supposed to make. Panel legend does not explain gray colored symbols (only black colored symbols)

Figure 3: plots carbon stabilization (parameterized as fraction of mineral-associated organic C per mass unit of soil) as a function of cumulated respiration (aka carbon loss), showing a positive correlation.

Figure 4: Gives the proportions of "unprotected C"; "biomass C", "mineral-associated C" and respired carbon as a function of experimental treatment, with

Panel A illustrating effect of root exudates against unamended control (with a lower proportion of mineral associated C in the control),

Panel B illustrating effects of variations in clay reactivity, suggesting greater association with minerals in systems having higher respiration and

Panel C illustrating the effects of substrate chemistry, showing a stimulation of respiration but no effect on proportion of mineral associated C

As with Figure 2, figure caption contains phraseology that does not convey actual information.

For instance: 403 "Post-hoc letters refer to an analysis of variance on total carbon pools in each microcosm at the time of harvest (i.e. the sum of unprotected, mineral-associated and microbial biomass carbon pools)"

What is a post-hoc letter? A letter that was assigned "after the fact"? Why use a language (latin) other than english ? Are you trying to demonstrate your superior education or are you just copying something you seen elsewhere ? Either way, you are not putting yourself in a favorable light when you substitute fashionable jargon for clear, precise and direct language.

What does it mean when one column is labeled with an 'a' and its neighbor with a 'b'? Does it mean the authors have performed an analysis of variance? Or is the letter supposed to tell the reader something about the result of this analysis of variance? If so, what does it tell the reader?

Conclusions

#1 75 the formation of persistent (mineral-associated) SOC is controlled by the interaction between mineral surfaces and roots

#2 77 the chemistry of C inputs did not impact SOC stabilization or the composition of microbial communities.

#3 79 soil C stabilization was positively correlated with C loss

Assessment

The observations reported in this manuscript provide experimental evidence for some suspicions that many in the scientific community have nurtured for some time. In that regard, the piece is novel, potentially transformative and definitely of interest to the scientific community.

That said, there are significant issues that need attention prior to publication:

a) This experiment was extremely complex. It has multiple temporal phases, artificial roots, artificial soils, real soils, variations in microbial community composition, treatments varying exudate application, clay mineralogy and substrate chemistry. The authors owe the reader a short narrative explaining the rationale behind such a complex setup. Why did you choose those treatments/time steps/manipulations? This is best done in some sort of flow chart listing the respective experimental activity together with the insights expected from that experimental manipulation.

b) Conclusion #1 = "the formation of persistent (mineral-associated) SOC is controlled by the interaction between mineral surfaces and roots"

is problematic on several levels. The authors equate mineral associated C with persistent C, without giving a quantitative definition of "persistence". They omit this definition probably because there is not a good one - "persistence" is not a measurable, quantifiable trait, it is merely a semantic prosthesis for something that is difficult to say otherwise. What the authors mean to say is that they believe that once associated with mineral surfaces, organic matter will automatically and reliably exhibit slower turnover. Unfortunately, especially when Fe oxides are involved, as in one of their mineralogy treatments, the opposite can happen - organic matter turnover can be accelerated when organic matter gets close to minerals.

Compare, for instance: Chen, C., Hall, S.J., Coward, E. et al. Iron-mediated organic matter decomposition in humid soils can counteract protection. *Nat Commun* 11, 2255 (2020).

This does not invalidate the widely known fact that greater proportions of highly reactive clays tend to promote C accumulation in soil - more clay may, for instance, lead to architectural features in the soil that are conducive to heterogenic (faster/slower) C turnover dynamics. Since factors such as soil structure were not investigated in this study, it is not possible to attribute the "formation of persistent C" exclusively to the interaction between minerals and roots. The only thing that can defensibly be claimed is that the size of the mineral associated carbon pool will respond to the abundance of co-metabolites aka root exudates, and this is actually a neat and complementary extension of the findings of :

Klotzbücher, T., Kaiser, K., Guggenberger, G., Gatzek, C. and Kalbitz, K., 2011. A new conceptual model for the fate of lignin in decomposing plant litter. *Ecology*, 92(5), pp.1052-1062.

who showed that cometabolites are required for the decomposition of Lignin.

Accordingly, what this work is corroborating is that plants (through exudation) can counterbalance mineral protection of organic substrates and so stimulate microbial activity in the rhizosphere as per ref 17. But the response observed does not allow to isolate root exudation as a single value predictor for carbon turnover rate.

c) The authors dwell in jargon. The manuscript is full of poorly defined, unmeasurable, non quantitative language, dwelling in terms such as "labile C"; "stable C"; "stabilization"; "C that resists"; "recalcitrant inputs"; "persistent C".

This practice dilutes the scientific edge of the piece and takes away from its potential impact. If the authors mean to refer to carbon that is cycling more slowly than other carbon, they should say so and identify the state against which the comparison of a turnover rate has been made. If they want to refer to carbon that may be physically protected because of its proximity to / encapsulation within mineral surfaces/aggregates, they should say so. Calling everything that is mineral associated "persistent" is not permissible, because of the rapidly increasing evidence for mineral surfaces as powerful agents of OM fragmentation.

Recommendation

This research is of interest to the scientific community and responds to several ongoing debates with the potential to largely resolve them. However, the way this MS is currently organized

prevents the reader from appreciating its potential impact. To remedy this, the authors are encouraged to rewrite the introduction around the following points:

1. Which overarching problem does this MS address?
2. What is/are the specific research question/s chosen for this investigation ?
3. Which scientific unknowns needed to be resolved to answer the specific research question?
4. Which assumptions/hypotheses needed to be tested to get at the unknowns?
5. Which strategy underlies the experimental design (= conceptual approach statement). Given the complexity of the experimental design, a flow chart explaining the relation between experimental activity and expected outcome will help the reader to understand the authors rationale and will help the authors to streamline their arguments

The results/discussions chapter should be reorganized to reflect the conceptual approach mentioned above. It is currently a mixture between philosophical treatise and factual report, and as such very difficult to read and digest. Results should verbalise observations made (Figure 2!!!) and the discussion part should provide an (i) internal plausibility check as well as a (ii) comparison with the existing literature.

Reviewer #2 (Remarks to the Author):

The main scientific question of the manuscript "What controls the size and persistence of the SOC pool" is still very relevant and worth to investigate because our quantitative knowledge about the main controls of the formation and stability of soil organic carbon (SOC) is fragmentary and partly contradictory. It is well-known that "C is both stabilized and lost from the system simultaneously." The authors designed a microcosm study using artificial roots to assess both SOC stabilization and losses. They used an innovative system of hollow fibre dialysis membranes to mimic the plant roots. Furthermore, they combined that with a variation in soil mineralogy, above ground litter input and soil microbial communities. Such a comprehensive approach comprising main potential controls of soil organic matter stabilization and decomposition is highly appreciated. Nevertheless, I have great concerns related to the manuscript because of many weaknesses:

The impression given by the authors that stabilization and losses of SOC were not studied together in the past is not completely true. It is well known that a high microbial activity (meaning large C mineralization) often results in an efficient formation of mineral-associated organic matter (direct pathway). As indicated above, that does not mean these processes are fully understood. However, a more hypothesis-driven study would be more appropriate.

Even more critically, the data presented by the authors are not convincing to support such a close relationship between stabilization and losses. For instance, a linear regression that explains 13% of the variability in the data (Fig. 3) is not very convincing in this respect. How relevant is a relationship explaining 4% of the data variability as the correlation between microbial biomass and mineral-associated carbon. Furthermore, I have some doubts regarding the microbial biomass data. Usually, microbial biomass C is less than 5% of total SOC, often around 1%. In this study, microbial biomass C is about 20% (or even more) of SOC. How realistic might that be for real soils? There is no comment / explanation / discussion at all related to these exceptional high values. The authors used a real soil in comparison to the artificial soils. But even in this soil, the microbial biomass C is very high. The very high addition of organic matter during the experiment might be one reason for that.

As a crucial treatment, the authors used three different compositions of the mineral assemblage in the artificial soils. In addition to kaolinite and montmorillonite, they used a treatment with montmorillonite and goethite (10 % of the clay mineral was replaced by the iron oxide/hydroxide). In the literature, montmorillonite is a high activity clay. The authors did not even mention that. They used a combination of a 2:1 clay mineral with a high specific surface area and a high cation exchange capacity (CEC; i.e. high activity clay) and an iron oxide/hydroxide. They called that mixture of a clay mineral and an iron oxide/hydroxide a high activity clay which is wrong and not an appropriate term. Even more critical is that the results were not linked to the different properties of the minerals. There is no discussion at all about potential effects of the different mineral surfaces on SOC stabilization and mineralization although they studied the mineral composition as an important experimental treatment. Furthermore, the reported CEC of the used

goethite is extraordinary high assuming a point of zero charge around 8.

In the manuscript, the authors assumed that mineral-associated organic matter is persistent. That is not necessarily the case. It is also not surprising and not very new that the proportion of a quite stable fraction does not dictate the pool size of total SOC.

In summary, I recommend to reject the manuscript because of scientific reasons: The main scientific question is not completely new as the authors claimed, the data are not convincing and there is no explanation of the data based on the different properties of the used minerals. As I stated above the experiment is very valuable. Therefore, the authors should use this approach and the obtained data to really go into the details of interactions between soil minerals, above- and belowground C input and their consequences for C turnover and the formation of mineral-associated organic matter.

We thank the Editor and two reviewers for their thoughtful comments on our work. We have conducted a substantial revision of our manuscript to address their critiques. These major revisions include:

- 1) Restructuring the entire introduction section to present our research questions and specific hypotheses, along with an experimental flowchart
- 2) Replacing vague terminology such as 'persistent' soil carbon or 'high-activity minerals' throughout
- 3) Including a section to consider how different organo-mineral bonding mechanisms on kaolinite, montmorillonite, and goethite might have influenced our results
- 4) Restructuring the results and discussion in line with our hypotheses

Below, we provide point-by-point responses to each of the Reviewers' suggestions, highlighting our responses in blue text for clarity.

REVIEWER COMMENTS

Reviewer #1 (Remarks to the Author):

MANUSCRIPT BRIEF

General Problem

26: What controls the size and persistence of the SOC pool?

Specific Research Question

15: What is the net impact of microbial metabolism on the total quantity of C held in the soil?

Scientific Unknown

59: We lack a quantitative framework to predict how a change in root production will impact SOC stocks

70: when labile C inputs accelerate C stabilization and loss simultaneously, there are uncertain consequences for the total C stock

Objectives

The scientific objectives pursued with this study were not identified

Hypotheses

The authors do not identify the hypotheses they were presumably testing. They must have assumed that variations in root influence, microbial community composition, mineralogy and substrate chemistry would lead to differential outcomes. It is not clear why they choose not to share their assumptions = hypotheses.

If the study was meant to be purely exploratory than this should be stated and rationalized.

We have substantially revised the introduction to describe our overall research aim, three principal research questions, and hypotheses. More detailed responses regarding this point can be found in our response to the Reviewer's 'Assessment,' below

Conceptual Approach

The only statement giving some level of strategic reasoning is found in the abstract:

16: we constructed artificial root-soil systems to identify controls on C cycling across the plant-microbe-mineral continuum, simultaneously quantifying the formation of persistent mineral-associated C and C losses to respiration. There is no rationale / justification / deduction explaining the choice of the three experimental treatments (root presence; mineralogy and microbial community composition)

The latter two factors are labeled as "soil forming" (94). The authors are invited to review contemporary reviews of soil forming factors to learn that both, mineralogy and microbial community composition are the result of pedogenesis and not its cause.

We have rephrased this research question to avoid the implication that microbes or the presence of specific minerals drives pedogenesis. On **Line 81-83**, we write

"... which is the principal driver of soil C balance: microbes, which drive SOC loss through mineralization, or minerals, which protect C from microbial attack?"

A description of the rather complex experimental setup follows 83-91, but rationalization does not go beyond 87 “to explore the effect on.....”

Results

Figure 1: illustrates influence of root exudates on

- cumulative C fluxes (decreasing with increasing clay activity)
- microbial biomass (increasing with increasing clay activity) and
- Mineral associated C pools (increasing with increasing clay activity)

Figure 2: Has two panels, one for bacteria and one for fungi. The caption is incomplete/inadequate in that it identifies the statistic procedure performed, but omits any hint at what point the panels are supposed to make. Panel legend does not explain gray colored symbols (only black colored symbols)

We have revised the caption as follows:

Figure 3. Non-metric multidimensional scaling analysis conducted on a Bray-Curtis dissimilarity matrix of both bacterial and fungal communities in the second phase of the experiment, demonstrating the strong impact of the microbial inoculum treatment (black vs. gray symbols), mineral reactivity (indicated by symbol shape), and root exudation (open/closed symbols) on bacterial and fungal assemblages. Other treatments and treatment interactions explained < 5% of compositional variance and are not shown here.

Figure 3: plots carbon stabilization (parameterized as fraction of mineral-associated organic C per mass unit of soil) as a function of cumulated respiration (aka carbon loss), showing a positive correlation.

Figure 4: Gives the proportions of “unprotected C”; “biomass C”, “mineral-associated C” and respired carbon as a function of experimental treatment, with

Panel A illustrating effect of root exudates against unamended control (with a lower proportion of mineral associated C in the control),

Panel B illustrating effects of variations in clay reactivity, suggesting greater association with minerals in systems having higher respiration and

Panel C illustrating the effects of substrate chemistry, showing a stimulation of respiration but no effect on proportion of mineral associated C

As with Figure 2, figure caption contains phraseology that does not convey actual information.

For instance: 403 “Post-hoc letters refer to an analysis of variance on total carbon pools in each microcosm at the time of harvest (i.e. the sum of unprotected, mineral-associated and microbial biomass carbon pools)”

What is a post-hoc letter? A letter that was assigned “after the fact”? Why use a language (latin) other than english ? Are you trying to demonstrate your superior education or are you just copying something you seen elsewhere ? Either way, you are not putting yourself in a favorable light when you substitute fashionable jargon for clear, precise and direct language. What does it mean when one column is labeled with an ‘a’ and its neighbor with a ‘b’? Does it mean the authors have performed an analysis of variance? Or is the letter supposed to tell the reader something about the result of this analysis of variance? If so, what does it tell the reader?

For clarity, we have revised the caption as follows:

Figure 5. Distribution of C pools in each microcosm as a function of experimental treatment: **A.** root exudates, **B.** mineral reactivity, and **C.** aboveground C amendment. All data shown are for artificial soils in the second phase of the experiment. All pools were directly and independently quantified with the exception of the unprotected (non-mineral associated) particulate organic carbon pool, which was determined via a mass balance approach. Error bars reflect standard errors; for the

unprotected C pool, these errors are propagated across all directly measured pools. Letters on the graphs indicate significant differences among treatment levels in Tukey's post-hoc tests, conducted following analysis of variance.

Conclusions

#1 75 the formation of persistent (mineral-associated) SOC is controlled by the interaction between mineral surfaces and roots

#2 77 the chemistry of C inputs did not impact SOC stabilization or the composition of microbial communities.

#3 79 soil C stabilization was positively correlated with C loss

Assessment

The observations reported in this manuscript provide experimental evidence for some suspicions that many in the scientific community have nurtured for some time. In that regard, the piece is novel, potentially transformative and definitely of interest to the scientific community. That said, there are significant issues that need attention prior to publication:

Thank you for these kind comments; below we detail our response to each of the reviewer's critiques

a) This experiment was extremely complex. It has multiple temporal phases, artificial roots, artificial soils, real soils, variations in microbial community composition, treatments varying exudate application, clay mineralogy and substrate chemistry. The authors owe the reader a short narrative explaining the rationale behind such a complex setup. Why did you choose those treatments/time steps/manipulations? This is best done in some sort of flow chart listing the respective experimental activity together with the insights expected from that experimental manipulation.

We have addressed this issue by providing a new Figure, which shows how each phase of the experiment maps onto our specific research questions and hypotheses. Additionally, we articulate these much more clearly on **Lines 78-105**:

Our goal was to evaluate the relative importance of plant, microbial, and mineralogical controls on two key processes that determine SOC pool size: mineral stabilization of new inputs, and respiratory soil C loss. To do so, we designed artificial root-soil systems to test three research questions:

First, which is the principal driver of soil C balance: microbes, which drive SOC loss through mineralization, or minerals, which protect C from microbial attack?

We hypothesized (H1) that mineral reactivity would exert the dominant control, both directly (through organic matter sorption) and indirectly (through its influence on microbial community dynamics) [...]

For our second question we asked, how do the chemistry and point-of-entry of C inputs (i.e. soil surface vs rhizosphere C additions) affect their interactions with microbes and minerals and thus, the subsequent fate of those inputs in the soil? [...]

We hypothesized that, in line with current conceptual frameworks, the most labile C inputs would be preferentially incorporated into microbial products (H2a), resulting in a larger biomass and mineral-associated C pool, especially in the presence of reactive minerals.

We also expected simple C in the form of root exudates to prime the decomposition and subsequent mineralization of more chemically complex inputs (H2b).

For our third and final question, we asked, what are the relationships among C stabilization, C mineralization, and soil C balance? [...]

We hypothesized (H3) that across all experimental treatments, C stabilization would be negatively correlated with C loss through respiration, because organic matter captured in the slower-cycling mineral-associated C pool would be protected from mineralization.

b) Conclusion #1 = “the formation of persistent (mineral-associated) SOC is controlled by the interaction between mineral surfaces and roots” is problematic on several levels. The authors equate mineral associated C with persistent C, without giving a quantitative definition of “persistence”. They omit this definition probably because there is not a good one - “persistence” is not a measurable, quantifiable trait, it is merely a semantic prosthesis for something that is difficult to say otherwise. What the authors mean to say is that they believe that once associated with mineral surfaces, organic matter will automatically and reliably exhibit slower turnover. Unfortunately, especially when Fe oxides are involved, as in one of their mineralogy treatments, the opposite can happen – organic matter turnover can be accelerated when organic matter gets close to minerals.

Compare, for instance: Chen, C., Hall, S.J., Coward, E. et al. Iron-mediated organic matter decomposition in humid soils can counteract protection. *Nat Commun* 11, 2255 (2020).

This raises an excellent point which we neglected to discuss in our previous draft – although on average mineral-associated C turns over much more slowly than particulate C, portions of it are highly dynamic and (as in the cited reference) metal oxides can promote C mineralization under anoxic conditions. To acknowledge this complexity, we have removed the term ‘persistence’ throughout the manuscript in favour of ‘slowly cycling.’ We define explicitly what we mean by carbon ‘stabilization’ on **Line 99-102**, meanwhile acknowledging that mineral-associated organic matter plays a complex role in carbon cycling:

Here, we define C ‘stabilization’ as the formation of mineral-associated C. Although under some circumstances minerals accelerate soil C loss (Chen et al. 2020), on average mineral-associated organic matter turns over up to 1000 times more slowly than particulate organic matter. (Georgiou et al. 2022)

Later on, we again evaluate the possibility that the presence of metal oxides accelerated C turnover in our artificial soils (**Lines 166-169**):

Moreover, both CO₂ fluxes and mineral-associated C were greatest in soils containing goethite (**Fig S6**). This suggests that metal oxides accelerated C stabilization and loss simultaneously, promoting the formation of organo-mineral bonds, but perhaps also coupling iron reduction with C mineralization in anoxic soil microsites. (Chen et al. 2020)

This does not invalidate the widely known fact that greater proportions of highly reactive clays tend to promote C accumulation in soil – more clay may, for instance, lead to architectural features in the soil that are conducive to heterogenic (faster/slower) C turnover dynamics. Since factors such as soil structure were not investigated in this study, it is not possible to attribute the “formation of persistent C” exclusively to the interaction between minerals and roots. The only thing that can defensibly be claimed is that the size of the mineral associated carbon pool will respond to the abundance of co-metabolites aka root exudates, and this is actually a neat and complementary extension of the findings of : Klotzbücher, T., Kaiser, K., Guggenberger, G., Gatzek, C. and Kalbitz, K., 2011. A new conceptual model for the fate of lignin in decomposing plant litter. *Ecology*, 92(5), pp.1052-1062. who showed that co-metabolites are required for the decomposition of Lignin. Accordingly, what this work is corroborating is that plants (through exudation) can counterbalance mineral protection of organic substrates and so stimulate microbial activity in the rhizosphere as per ref 17. But the response observed does not allow to isolate root exudation as a single value predictor for carbon turnover rate.

We do not claim that root exudation is the sole or even most important predictor of carbon turnover. For example, on **Line 211-213** we write:

Taken together, our data illustrate how soil mineralogy mediates the effect of roots on C stabilization and loss mechanisms, shaping both the size and chemical composition of the SOC pool.

And on **Lines 236-238**:

When viewed holistically, data from our experiment suggest that roots, microbes, and minerals interact to regulate both soil C loss via respiration and stabilization via mineral-associated C formation.

However, the Reviewer's comments demonstrate that the last version of our manuscript was not sufficiently clear about the nuanced role of root exudates in mediating carbon cycling, or how root effects were modified by minerals. We have included a new paragraph (**Lines 211-235**) discussing this.

c) The authors dwell in jargon. The manuscript is full of poorly defined, unmeasurable, non quantitative language, dwelling in terms such as "labile C"; "stable C"; "stabilization"; "C that resists"; "recalcitrant inputs"; "persistent C".

This practice dilutes the scientific edge of the piece and takes away from its potential impact. If the authors mean to refer to carbon that is cycling more slowly than other carbon, they should say so and identify the state against which the comparison of a turnover rate has been made. If they want to refer to carbon that may be physically protected because of its proximity to / encapsulation within mineral surfaces/aggregates, they should say so. Calling everything that is mineral associated "persistent" is not permissible, because of the rapidly increasing evidence for mineral surfaces as powerful agents of OM fragmentation.

We have removed the term 'persistent' and replaced it with 'slow cycling,' and eliminated 'recalcitrant' in favour of 'chemically complex.' We continue to use the term stabilization, but only after explicitly defining it with relation to the pools and fluxes measured (e.g. **Line 99**).

Recommendation

This research is of interest to the scientific community and responds to several ongoing debates with the potential to largely resolve them. However, the way this MS is currently organized prevents the reader from appreciating its potential impact. To remedy this, the authors are encouraged to rewrite the introduction around the following points:

1. Which overarching problem does this MS address?
 2. What is/are the specific research question/s chosen for this investigation ?
 3. Which scientific unknowns needed to be resolved to answer the specific research question?
 4. Which assumptions/hypotheses needed to be tested to get at the unknowns?
 5. Which strategy underlies the experimental design (= conceptual approach statement).
- Given the complexity of the experimental design, a flow chart explaining the relation between experimental activity and expected outcome will help the reader to understand the authors rationale and will help the authors to streamline their arguments

We hope the inclusion of the new **Figure 1**, along with our major revisions detailed above, clarify each of these points.

The results/discussions chapter should be reorganized to reflect the conceptual approach mentioned above. It is currently a mixture between philosophical treatise and factual report, and as such very difficult to read and digest. Results should verbalise observations made (Figure 2!!!) and the discussion part should provide an (i) internal plausibility check as well as a (ii) comparison with the existing literature.

We have re-organized the discussion such that each of the hypotheses articulated in the introduction is evaluated in a separate paragraph, with the following paragraph used to explore the biological mechanisms underlying each observed pattern. Below, we have reproduced each of the hypothesis evaluations in the results and discussion section:

Contrary to our hypothesis (H1), during the first three months mineral reactivity did not influence soil C balance, but the total C pool in each microcosm (calculated via a mass-balance approach) was 4% larger in the FB vs. BO treatment (**Table S1**). This is because microcosms in the BO treatment exhibited 20% greater rates of respiration (**Table S1**). (**Lines 125-130**)

We expected to find an overall trend towards lower respiration, microbial biomass and mineral-associated C in the xylan treatment reflecting slower decomposition and less efficient conversion of plant biomolecules into microbial tissue (H2a). However, these predictions were only partially supported. While respiration rates were 41% lower in microcosms amended with xylan vs. cellobiose or glucose (**Fig S6**), the microbial biomass was actually slightly *larger* in the xylan treatment than in the glucose treatment (**Fig S6**), and aboveground C input chemistry had no impact on mineral-associated C formation (**Table S3, Fig S6**). (**Lines 170-176**)

We anticipated strong interactions between the root treatment and the composition of non-root C inputs, with exudates accelerating decomposition of the complex polymer xylan, but not glucose or cellobiose - i.e., the priming effect (H2b). This hypothesis was also unsupported, as interactions between C input chemistry and root exudation treatments were generally insignificant for biogeochemical variables (**Table S3**). (**Lines 190-193**)

These processes are often assumed – implicitly or explicitly – to be negatively correlated, such that an increase in soil C loss by respiration will reduce the availability of organic matter for stabilization (or, conversely, that C capture in mineral-associated pools reduces the amount of substrate available for mineralization) (H3) [...] We explored the relationship between these two processes using a linear regression, controlling for the total quantity of C inputs to each microcosm in each time interval. We found that C losses (cumulative respiration) were strongly *positively* correlated with C stabilization (mineral-associated C stocks) (**Lines 238-249**)

We also provide 'plausibility checks' by evaluating artificial soil C pools against empirical data (**Lines 154-161**).

Reviewer #2 (Remarks to the Author):

The main scientific question of the manuscript "What controls the size and persistence of the SOC pool" is still very relevant and worth to investigate because our quantitative knowledge about the main controls of the formation and stability of soil organic carbon (SOC) is fragmentary and partly contradictory. It is well-known that "C is both stabilized and lost from the system simultaneously." The authors designed a microcosm study using artificial roots to assess both SOC stabilization and losses. They used an innovative system of hollow fibre dialysis membranes to mimic the plant roots. Furthermore, they combined that with a variation in soil mineralogy, above ground litter input and soil microbial communities. Such a comprehensive approach comprising main potential controls of soil organic matter stabilization and decomposition is highly appreciated. Nevertheless, I have great concerns related to the manuscript because of many weaknesses:

Thank you for your evaluation – we have addressed each of the weaknesses described by the reviewer, as discussed below.

The impression given by the authors that stabilization and losses of SOC were not studied together in the past is not completely true. It is well known that a high microbial activity (meaning large C mineralization) often results in an efficient formation of mineral-associated organic matter (direct pathway). As indicated above, that does not mean these processes are fully understood. However, a more hypothesis-driven study would be more appropriate.

As discussed in our response to Reviewer 1, we have substantially revised the introduction to provide our overall experimental aim, the three research questions that motivated the study, and the associated hypotheses. We return to these hypotheses in the results and discussion to evaluate the evidence for each.

Even more critically, the data presented by the authors are not convincing to support such a close relationship between stabilization and losses. For instance, a linear regression that explains 13% of the variability in the data (Fig. 3) is not very convincing in this respect.

What is surprising about the relationship shown here is not the magnitude of the effect but rather its sign – there is a significant, **positive** relationship between soil C stabilization and C loss. As we articulate on **Line 102-105**,

We hypothesized (H3) that across all experimental treatments, C stabilization would be **negatively** correlated with C loss through respiration, because organic matter captured in the slower-cycling mineral-associated C pool would be protected from mineralization.

As we discuss in more detail in response to one of the Reviewer's subsequent points below, the assumption that soil carbon losses can be minimized (and soil stocks enhanced) via the formation of 'stable' carbon is quite common in the literature. Our findings are therefore quite valuable, because they illustrate that these assumptions are frequently invalid.

How relevant is a relationship explaining 4% of the data variability as the correlation between microbial biomass and mineral-associated carbon.

We have removed this analysis from the manuscript

Furthermore, I have some doubts regarding the microbial biomass data. Usually, microbial biomass C is less than 5% of total SOC, often around 1%. In this study, microbial biomass C is about 20% (or even more) of SOC. How realistic might that be for real soils? There is no comment / explanation / discussion at all related to these exceptional high values. The authors used a real soil in comparison to the artificial soils. But even in this soil, the microbial biomass C is very high. The very high addition of organic matter during the experiment might be one reason for that.

We agree, and have added the following information to the manuscript (**Lines 158-161**):

The microbial biomass accounted for a comparatively large fraction (30.8%) of the total SOC pool. This likely reflects the high rate of C inputs, as the figure was similar (16.3%) in the real-soil controls.

The above figures express the quantity of C in the biomass as a fraction of the quantity of C held in the soil at time of harvest. We can also calculate a time-integrated carbon use efficiency for the microbial biomass as: $CUE = \frac{\text{biomass C}}{\text{biomass C} + \text{respired C}}$ to explore the fraction of available carbon immobilized in the biomass vs. respired. (This calculation is distinct from the instantaneous CUE measured via isotope probing at the end of the first harvest). We obtain a CUE of 0.194 ± 0.010 for artificial soils, and 0.184 ± 0.035 for artificial soils. These values are somewhat less than the global mean of 0.30 suggested by (Sinsabaugh et al.,

2013), but suggest that the size of the biomass is reasonably proportionate to the total amount of carbon added, given well-understood constraints on microbial metabolism.

As a crucial treatment, the authors used three different compositions of the mineral assemblage in the artificial soils. In addition to kaolinite and montmorillonite, they used a treatment with montmorillonite and goethite (10 % of the clay mineral was replaced by the iron oxide/hydroxide). In the literature, montmorillonite is a high activity clay. The authors did not even mention that.

We provide more detail about the mineral treatments, discussing the effects of mineral surface area, CEC, and charge, on **Lines 211-235** (as discussed further below.)

They used a combination of a 2:1 clay mineral with a high specific surface area and a high cation exchange capacity (CEC; i.e. high activity clay) and an iron oxide/hydroxide. They called that mixture of a clay mineral and an iron oxide/hydroxide a high activity clay which is wrong and not an appropriate term.

We have replaced the term 'clay activity' with 'mineral reactivity' throughout, to reflect the fact that one of our treatment levels contained metal oxides in addition to clays.

Even more critical is that the results were not linked to the different properties of the minerals. There is no discussion at all about potential effects of the different mineral surfaces on SOC stabilization and mineralization although they studied the mineral composition as an important experimental treatment.

This is a great point and we have provided a new paragraph to more extensively consider the role of specific mineral properties in influencing our results (**Lines 211-235**):

Soils containing goethite exhibited larger mineral-associated C pools than those consisting of clay minerals only, demonstrating the superior C stabilization capacity of metal oxides in comparison with phyllosilicate clays across a broad range of soil conditions. Moreover, only in the presence of goethite did roots enhance, rather than diminish, mineral-associated C formation. These patterns may shed light on the mineral traits that govern the stabilization of root exudates. Kaolinite and montmorillonite had very similar impacts on mineral-associated C formation, despite their very different surface areas and cation exchange capacities; this suggests that neither CEC nor clay surface area strongly influenced the fate of C inputs in our artificial soils. However, both phyllosilicate minerals differ from goethite in a critical respect: in contrast with 1:1 phyllosilicate minerals like kaolinite (usually uncharged) and 2:1 phyllosilicate minerals like montmorillonite with a permanent negative charge, metal oxides like goethite exhibit variable charge and bear highly reactive hydroxyl groups. Bonds formed through ligand exchange between the hydroxyl groups of organic matter and metal oxides are among the most durable, and explain the particular affinity of organic acids and other hydroxyl/carboxyl-rich compounds for goethite. Although the simulated root exudate solution we added did not contain these compounds, it did provide simple sugars that fuel microbial growth and production of carboxyl/hydroxyl-rich metabolites, e.g. oxalic acid or glutamate. By contrast, the compounds most likely to be stabilized on negatively charged phyllosilicate clays, primarily via ionic bonds, would include basic amino acids (which were included in the simulated root exudate), or microbial residues stabilized via cation bridging. Both of these bonding mechanisms are weaker than the ligand exchange observed on goethite. Therefore, we suggest that root-induced acceleration of microbial growth promoted stabilization of biomass residues on metal oxide surfaces, but disrupted weaker organo-mineral bonds in phyllosilicate clays.

Furthermore, the reported CEC of the used goethite is extraordinary high assuming a point of zero charge around 8.

We were also surprised by the very high CEC of goethite. This CEC measurement suggests that goethite carried a negative charge, which in turn implies that the soils could have been relatively alkaline. However, we measured CEC on the pure minerals, not the artificial soils containing the goethite, which may have been more acidic overall due to the organic matter content. Regardless of the charge state of the goethite, our arguments about its superior organic matter stabilization capacity would be unaffected. Prior work demonstrates that goethite bears reactive hydroxyl groups which can strongly stabilize organic matter even at high pH, when some of the hydroxyls are deprotonated (Kahle et al., 2004). Organic matter sorption onto goethite is greater than for kaolinite and montmorillonite across the entire soil pH range from 3 to 10 (Roychand et al., 2010).

In the manuscript, the authors assumed that mineral-associated organic matter is persistent. That is not necessarily the case.

Good point; as discussed in our response to Reviewer 1 above, we have acknowledged the complexity of mineral-associated C pools on **Lines 100-103**, and acknowledged that goethite may have accelerated C mineralization on **Lines 167-169**.

It is also not surprising and not very new that the proportion of a quite stable fraction does not dictate the pool size of total SOC.

The reviewer is correct that this point is not new (for instance, see the excellent opinion piece by (Janzen, 2006)). However, we argue that it is definitely worth repeating, and demonstrating with empirical data (the novel advance of our contribution). The idea that 'soil carbon stabilization' is equivalent to 'soil carbon sequestration' is still quite pervasive in the literature. Below we have provided a few examples (noting that the papers from which we quote all represent high-quality scientific contributions; we are not criticizing the authors). The assumption that 'stable' soil carbon is somehow more relevant to climate mitigation efforts dictates not only our community's research priorities, but also the management of soils for carbon sequestration. We therefore feel that our paper makes an important and timely contribution to this dialogue.

"Improved soil management practices that promote soil carbon sequestration, especially in stable carbon pools, are needed to reverse this trajectory and mitigate climate change... mineral-associated organic carbon (MOC) can have turnover times up to 1000 times longer (reaching 100-10,000 years) than particulate organic carbon at the same depth. Thus, increasing MOC may be a key to lasting carbon sequestration in soils." (Georgiou et al., 2022)

"The combination of strong physico-chemical sorption and spatial separation from microbial decay through protective aggregates means MAOM turns over much more slowly than POM [particulate organic matter] and can remain in soils for long periods of time. From a climate change mitigation perspective, increasing soil C content of the MAOM fraction is more desirable than increasing the POM fraction, since this represents a more stable long-term C reservoir in soils." (Midwood et al., 2021)

"To increase C sequestration in soil, elevated CO₂ must increase C pools with turnover times of decades and centuries. C in the active pool is lost too quickly." (Beedlow et al., 2004)

In summary, I recommend to reject the manuscript because of scientific reasons: The main scientific question is not completely new as the authors claimed, the data are not convincing and there is no explanation of the data based on the different properties of the used

minerals. As I stated above the experiment is very valuable. Therefore, the authors should use this approach and the obtained data to really go into the details of interactions between soil minerals, above- and belowground C input and their consequences for C turnover and the formation of mineral-associated organic matter.

We hope that the revisions detailed above satisfy the reviewer's concerns on both points.

References cited in this response

- Beedlow, P. A., Tingey, D. T., Phillips, D. L., Hogsett, W. E., & Olszyk, D. M. (2004). Rising Atmospheric CO₂ and Carbon Sequestration in Forests. *Frontiers in Ecology and the Environment*, 2(6), 315. <https://doi.org/10.2307/3868407>
- Georgiou, K., Jackson, R. B., Vindušková, O., Abramoff, R. Z., Ahlström, A., Feng, W., Harden, J. W., Pellegrini, A. F. A., Polley, H. W., Soong, J. L., Riley, W. J., & Torn, M. S. (2022). Global stocks and capacity of mineral-associated soil organic carbon. *Nature Communications*, 13(1), 3797. <https://doi.org/10.1038/s41467-022-31540-9>
- Janzen, H. H. (2006). The soil carbon dilemma: Shall we hoard it or use it? *Soil Biology and Biochemistry*, 38(3), 419–424. <https://doi.org/10.1016/j.soilbio.2005.10.008>
- Kahle, M., Kleber, M., & Jahn, R. (2004). Retention of dissolved organic matter by phyllosilicate and soil clay fractions in relation to mineral properties. *Organic Geochemistry*, 35(3), 269–276. <https://doi.org/10.1016/j.orggeochem.2003.11.008>
- Midwood, A. J., Hannam, K. D., Gebretsadikan, T., Emde, D., & Jones, M. D. (2021). Storage of soil carbon as particulate and mineral associated organic matter in irrigated woody perennial crops. *Geoderma*, 403, 115185. <https://doi.org/10.1016/j.geoderma.2021.115185>
- Roychand, P., Angove, M., & Tisdall, J. (2010). Sorptive protection of organic matter in soil. *World Congress of Soil Science, Soil Solutions for a Changing World, August*, 235–238.
- Sinsabaugh, R. L., Manzoni, S., Moorhead, D. L., & Richter, A. (2013). Carbon use efficiency of microbial communities: Stoichiometry, methodology and modelling. *Ecology Letters*, 16(7), 930–939. <https://doi.org/10.1111/ele.12113>

Reviewer #1 (Remarks to the Author):

The authors have satisfactorily addressed the concerns raised. They do however, maintain elements of jargon that perpetuate mechanistic falsehoods. These elements are the terms "resisting decomposition" and "labile C". Replacing these misleading terms with appropriate descriptive phraseology requires relatively simple rewording and should hence be corrected prior to publication. Please compare comments to authors below. The matter is serious but can be easily resolved and should hence be enforced.

Reviewer #3 (Remarks to the Author):

Summary of the study:

The authors conducted impressive work in designing an innovative experiment that tackles at the same time several factors that affect the cycling of C in soil, namely, (i) root exudates, (ii) aboveground C inputs, (iii) microbial community composition and (iii) soil mineralogy. The experimental approach consists of two phases. In the first phase, they evaluated the following factors: root exudates (with and without), microbial community composition (bacteria, fungi + bacteria), and mineral reactivity (Low, Medium, High). Some of the experimental unities were harvested and measured: soil C loss (cumulative CO₂), MAOM C, microbial analysis (microbial biomass, etc.). This step aims to evaluate the dominant control that governs soil C balance: mineral or soil. It is not mentioned in the hypothesis or in the scheme presented in fig. 1, but it refers to the C balance derived from the C input from the roots exudates (fig S9).

For the second phase, the remaining microcosms were incubated again. Another factor was included: aboveground C inputs with contrasting complexities. Concomitantly, they incubated real soil to validate their results. After 10 months, the samples were harvested, and the same analysis of the first phase was repeated. The aim of the second phase was to evaluate the effect of chemistry and point-of-entry of C inputs on soil C fate (L. 88). The authors present two hypotheses about this step:

H2a. They hypothesize that labile C (aboveground or root-exudates) would be preferentially incorporated into microbial products, resulting in higher microbial biomass and a higher MAOM, especially in the presence of reactive minerals.

H2b. The authors predict that simple C inputs in the form of root exudates would enhance the mineralization (CO₂) of more chemically complex inputs via priming effects. In sequence, the authors do a correlation between the MAOM C pool and soil respiration using the dataset of both phases. The hypothesis associated with this analysis: they expect to observe a negative correlation between C loss through respiration and MAOM formation. Instead of observing a negative correlation, Fig. 4 shows a positive correlation between cumulative CO₂ and MAOM formation.

Major comments:

As it is possible to observe, the described experimental design is really complex. It requires a great effort from the reader even to identify the factors that have been studied, and this is one of the greatest weaknesses of this study. I also have some concerns if the experimental approach that can be used to test the hypotheses that were proposed. In this context, I do not recommend the publication of the manuscript in the conditions that were submitted. Despite this, I would emphasize that the dataset generated by the authors is extremely valuable, and it can respond to several key questions about the C cycling in the soil. Bellow, I detailed my major concerns:

Introduction: I feel that it is a lack of connection between the introduction and the experimental approach. Some factors of the study are not properly developed, such as the impact of microbial community structure (bacteria vs. fungi). I would consider it more appropriate to enumerate all the factors you are trying to study and write a brief paragraph of each. Moreover, the hypotheses are presented without the rationale that was used to derive it. When a hypothesis is presented, the reader should be able to

identify the line of thought that the authors use to build it. Otherwise, the hypothesis appears to be senseless. This problem may be related to the large number of hypotheses that this study is trying to tackle at the same and to the complex two-step experimental approach. I would recommend keeping only relevant hypotheses to the goal of the study, as not all findings of the study need to be posed as a hypothesis.

Phase 1:

The goal of this step is to evaluate what the dominant control that governs soil C balance: mineral or microbial? They conclude that microbial community structure dominates mineral on C losses and C potentially (?) in C stabilization (L. 143). This conclusion is based on the following observation: no significant effect of Mineral reactivity for Cumulative CO₂ (p-value 0.080ns - Table S1).

I consider that the investigation of the relative importance of mineral and microbial factors to C stabilization would require a wider range of treatments that are not included in your experimental approach. Specifically, the microbial manipulation treatment was only limited to investigating differences between a population with only bacteria and another population with bacteria + fungi. In this context, the microbial control that was studied is limited to the presence/absence of fungi. The same can be said for the mineral factors, which were specifically addressed to investigate the effects of mineral activity. In this way, I do not consider that your experimental approach is suited for investigating what you proposed in the goal of this step. My recommendation here would be to restructure the goal/hypothesis of this phase, attaining specifically to what was tested in your experimental approach (mineral activity and fungi presence).

Phase 2:

The factor point-of-entry is not well explained in the introduction, and the description of the schemes is confusing. As far as I understood, the authors used different reactants with distinct complexity as aboveground C inputs that were applied at the top of the soil. Meanwhile, with the artificial root system, they applied a solution composed of a mixture of also labile compounds. In this case, you have microcosms that received C only from root exudates and microcosms that received C from exudates + C inputs from the top (aboveground C inputs). This means that the total amount of C applied in treatment with exudation and without exudation was distinct. Thus, there is an overlap between the factor "point of entry" and the amount of C entered into the soil. In this context, I do not consider that this experimental approach allows the investigation of the point-of-entry of C inputs (aboveground vs. belowground). To investigate that, you must have aboveground and belowground treatments with the same amount/type of C inputs (such as <https://doi.org/10.1038/s41561-018-0258-6>). Does your experimental design have this?

The H3 seems contradictory and needs to be better explained or even reconsidered. First, I would recommend changing the term "C stabilization" to "mineral-associated OM formation" to avoid misinterpretation and to explicit the exact measurement that was done. Second, the CO₂ production that was quantified is the sum of CO₂ decomposition from C inputs and the decomposition of preexistent SOM (formed in phase 1). It is possible that the negative relationship that you were investigating requires the separation between the CO₂ that was derived from the decomposition of the C inputs and the CO₂ emitted from preexisting SOM decomposition. Overall, I find that this result is not very surprising, and it is difficult to infer what exactly is happening without a more detailed approach that allows the separation of the sources of CO₂. I would recommend focusing on other, more interesting results in your study.

Minor comments:

Title: The title of MS, "Positive correlations between soil carbon stabilization and loss are regulated by roots, minerals and microbes" is not very much informative. The fact that roots, minerals and microbes regulate soil C stabilization is widely accepted and is possibly why the authors considered it as study factor. The only correlation that is present in the MS is related to MAOM C and the cumulative CO₂. In this context, the use of the plural, in this case, appears to be inappropriate.

Balance: during several passages, the authors refer to C balance. I implied they are

referring to the variables Total C pool and Cumulative CO₂ (table S1). This needs to be clearly defined. By the way, in table S1 the values of the Total C pool and cumulative CO₂ are similar. Is this a typo, or is it related to the mass balance approach that you used?

Labile C inputs: In several passages, the authors refer to labile C inputs, but it is rather unclear if they refer to aboveground (different complexities treatments) or to root exudates. One possible confusion is that root C inputs are also formed by labile C inputs (glucose, for example).

Schematic of the experimental approach: The two-step approach is really interesting and innovative, but it is really complicated to keep track of all the factors that were investigated in each step. Fig. 1 is helpful but needs to be improved. I would suggest combining Fig S9 with Fig. 1. Several aspects of the experimental design were only possible to understand after looking into the supplemental material. For example, in the second step, the experimental units received two types of C input (from the root exudates and also aboveground C inputs) at the same time.

Validation with real soil and plants: Despite the enormous importance of this for your study, I would keep the validation of the root exudates methods in the supplement materials and just mention it in the Methods section.

Figure 2: Is there any specific reason for presenting the Log response ratios instead of the original values? Also, are the differences between the mineral reactivity treatments significantly different from the control? What statistical test was used to determine this? A more detailed description of the statistics would be required. Also, what is the control that is being described here? Do you refer to the treatment that did not receive exudates?

Fig.5: We can observe that in most of the treatments, the cumulative C of C was respired. How did you separate the microbial biomass from the remaining SOM fractions? You measured the heavy fraction after the extraction of microbial biomass?
L. 117: Exchange the term dramatic.

L. 248: The term strong is vague. Also an R² of 0.131 shows that the losses of C are not explaining very much the variation in the MAOM formation.

Soil respiration: By reading L. 351., I understood that the C loss produced in each microcosm was not measured continuously. I would consider it essential to have more details of how this cumulative C loss was calculated; did you consider daily CO₂ rate uniform among the sampling dates?

Fig S1: Panel C: Lacks a description of the statistics used. Also, does "biomass" refer to "microbial biomass" or something else?

Fig S5. Lack of description of statistics again.

Table S2 and table S3: are analyses described in the statistics section?

We thank the editor and reviewers for the opportunity to revise our manuscript in line with their thoughtful comments and suggestions. Below, we respond point-by-point to each of the issue raised.

REVIEWER COMMENTS

Reviewer #1 (Remarks to the Author):

The authors have satisfactorily addressed the concerns raised. They do however, maintain elements of jargon that perpetuate mechanistic falsehoods. These elements are the terms "resisting decomposition" and "labile C". Replacing these misleading terms with appropriate descriptive phraseology requires relatively simple rewording and should hence be corrected prior to publication. Please compare comments to authors below. The matter is serious but can be easily resolved and should hence be enforced.

We have replaced both of these terms throughout the manuscript – for example:

- on **Line 34** we have replaced 'most C that resists decay' with 'most C with a long residence time.'
- On **Line 70-71** we replaced 'labile plant compounds' with 'the most bioavailable plant compounds (e.g. simple carbohydrates or amino acids)'

Reviewer #3 (Remarks to the Author):

Major comments:

As it is possible to observe, the described experimental design is really complex. It requires a great effort from the reader even to identify the factors that have been studied, and this is one of the greatest weaknesses of this study. I also have some concerns if the experimental approach that can be used to test the hypotheses that were proposed. In this context, I do not recommend the publication of the manuscript in the conditions that were submitted. Despite this, I would emphasize that the dataset generated by the authors is extremely valuable, and it can respond to several key questions about the C cycling in the soil. Bellow, I detailed my major concerns:

Thank you for your insightful comments, and we hope the revisions detailed below will permit the reader to better understand the design of the study and understand the justification for our hypotheses.

Introduction: I feel that it is a lack of connection between the introduction and the experimental approach. Some factors of the study are not properly developed, such as the impact of microbial community structure (bacteria vs. fungi). I would consider it more appropriate to enumerate all the factors you are trying to study and write a brief paragraph of each.

We have revised the introduction such that each of the four factors considered in our experiment (mineral reactivity, microbial community structure, presence of roots, and chemistry of non-root inputs) are introduced sequentially. In particular, we elaborate

on the roles of minerals and microbes in soil C dynamics in the paragraph beginning on **Line 50**:

... there are multiple interacting drivers of SOC cycling along the mineral-microbe-plant continuum that can influence microbial physiology. For example, chemical properties of the soil parent material dictate the capacity of soils to retain fresh C inputs, and shape microbial community structure as well. There is increasing recognition that clay mineralogy is a better predictor of SOC dynamics than overall clay content: at global scale, highly reactive minerals, such as high surface-area phyllosilicate clays and oxyhydroxides, are associated with larger mineral-associated SOC pools. By contrast, field-based and laboratory studies have found that microbial community structure has a larger impact on SOC dynamics than the chemistry of clay minerals, with fungi in particular playing a key role in the formation and turnover of particulate organic matter. These divergent findings may reflect the different scales at which drivers of SOC were evaluated (global vs. regional), or perhaps the identity of the dominant SOC pool in each system being examined (i.e., particulate vs. mineral-associated C). For example, C cycle dynamics in highly organic soils might be more sensitive to microbial community dynamics than clay mineralogy, while the reverse should be true for mineral soils.

Moreover, the hypotheses are presented without the rationale that was used to derive it. When a hypothesis is presented, the reader should be able to identify the line of thought that the authors use to build it. Otherwise, the hypothesis appears to be senseless. This problem may be related to the large number of hypotheses that this study is trying to tackle at the same and to the complex two-step experimental approach. I would recommend keeping only relevant hypotheses to the goal of the study, as not all findings of the study need to be posed as a hypothesis.

We hope that the major revisions to the introduction, together with the elaboration of our experimental design (described in more detail below) and refinement of the hypotheses themselves (also below), have resolved this issue.

Phase 1:

The goal of this step is to evaluate what the dominant control that governs soil C balance: mineral or microbial? They conclude that microbial community structure dominates mineral on C losses and C potentially (?) in C stabilization (L. 143). This conclusion is based on the following observation: no significant effect of Mineral reactivity for Cumulative CO₂ (p-value 0.080ns - Table S1).

I consider that the investigation of the relative importance of mineral and microbial factors to C stabilization would require a wider range of treatments that are not included in your experimental approach. Specifically, the microbial manipulation treatment was only limited to investigating differences between a population with only bacteria and another population with bacteria + fungi. In this context, the microbial control that was studied is limited to the presence/absence of fungi. The same can be said for the mineral factors, which were specifically addressed to investigate the effects of mineral activity. In this way, I do not consider that your experimental approach is suited for investigating what you proposed in the goal of this step. My recommendation here would be to restructure the goal/hypothesis of this phase, attaining specifically to what was tested in your experimental approach (mineral activity and fungi presence).

We hope our revised introduction (see paragraph beginning on **Line 50**) makes clear why we were specifically interested in the role of fungi vs. bacteria, and in clay mineralogy rather than soil clay content. We make this explicit on **Line 99-106**:

We hypothesized (H1) that, in the mineral soils we studied, clay mineralogy would exert the dominant control, both directly (through organic matter sorption) and indirectly (through its influence on microbial community dynamics). We evaluated this hypothesis by creating artificial soils in which we could independently manipulate microbial assemblages (inoculating with whole soil communities or with bacteria only) and mineral composition (keeping clay content constant across all the soils, but varying mineral reactivity; **Fig 1**).

We also note, however, that our experimental treatments affected microbial community composition and not just the fungal:bacterial ratio. This is illustrated in **Figure S5**, which shows that the inoculum treatments affected the diversity and taxonomic structure of both bacterial and fungal communities.

Phase 2:

The factor point-of-entry is not well explained in the introduction, and the description of the schemes is confusing. As far as I understood, the authors used different reactants with distinct complexity as aboveground C inputs that were applied at the top of the soil. Meanwhile, with the artificial root system, they applied a solution composed of a mixture of also labile compounds. In this case, you have microcosms that received C only from root exudates and microcosms that received C from exudates + C inputs from the top (aboveground C inputs). This means that the total amount of C applied in treatment with exudation and without exudation was distinct. Thus, there is an overlap between the factor "point of entry" and the amount of C entered into the soil. In this context, I do not consider that this experimental approach allows the investigation of the point-of-entry of C inputs (aboveground vs. belowground). To investigate that, you must have aboveground and belowground treatments with the same amount/type of C inputs (such as <https://doi.org/10.1038/s41561-018-0258-6>). Does your experimental design have this?

We apologize that our description of the root treatment was confusing, and agree with the reviewer's point. We cannot completely tease apart the effects of C input chemistry vs. point-of-entry, both because root exudates consisted only of simple C, and because microcosms with root exudates received more C in total than those without. These decisions were made intentionally. First, we did not introduce chemically complex C through the artificial roots, because exudates released *in situ* contain small bioavailable molecules and not complex polymers.¹ Therefore, it would be unrealistic and biologically irrelevant to conduct a fully factorial manipulation of the chemistry of aboveground C inputs vs. root exudates. Second, although microcosms in the root exudate treatment received more C in total, this was because, as we write on **Line 108-110**, 'We were particularly interested in whether the effect of root C inputs on overall SOC stocks is dependent upon the chemical complexity of non-root C inputs.' Our subsequent interpretation of the data takes the different amounts of C added to each microcosm into account, as described in more detail below.

To make our experimental design clearer, we have done the following:

- In the introduction, on **Line 81**, we have re-framed our discussion of the importance of root exudates in soil C cycling:

The fate of new C inputs to soil is determined not only by their chemical composition, but by their mechanism of delivery. Root exudates appear to play a particularly important role in SOC formation, for multiple reasons: exudates contain simple compounds that preferentially assimilated by microbes,¹⁹ and enter the soil in close proximity to the minerals that may ultimately stabilise these microbial residues²⁰ (i.e., the 'microbial carbon pump'). At the same time, however, the high bioavailability of the organic matter in root exudates can also accelerate microbial decomposition of unprotected C via the priming effect

- We provided a clearer description of the experimental design of Phase 2, both on **Line 113-115** ('Therefore, in its second phase, the experiment consisted of a fully factorial manipulation of clay mineralogy, initial microbial inoculum, the chemistry of surface C amendments (ranging from the monomer glucose to the complex polymer xylan), and root exudation (present or absent).') and in the new **Figure 1**
- In the discussion, we flag the fact that microcosms in the root exudate treatment received more C than those in the no-root controls (**Line 284-285**)

The H3 seems contradictory and needs to be better explained or even reconsidered.

We have rephrased this hypothesis on **Line 126-130**:

We hypothesized (H3) that across all experimental treatments, the formation of mineral-associated C would be negatively correlated with C loss through respiration. This is because organic matter captured in the slower-cycling mineral-associated C pool is protected from microbial attack, reducing the total quantity of C available for mineralization.

First, I would recommend changing the term "C stabilization" to "mineral-associated OM formation" to avoid misinterpretation and to explicit the exact measurement that was done.

We have done so throughout, and only retain the term stabilization when immediately followed with a description of what was measured (e.g. **Line 133**)

Second, the CO₂ production that was quantified is the sum of CO₂ decomposition from C inputs and the decomposition of preexistent SOM (formed in phase 1). It is possible that the negative relationship that you were investigating requires the separation between the CO₂ that was derived from the decomposition of the C inputs and the CO₂ emitted from preexisting SOM decomposition. Overall, I find that this result is not very surprising, and it is difficult to infer what exactly is happening without a more detailed approach that allows the separation of the sources of CO₂. I would recommend focusing on other, more interesting results in your study.

We have acknowledged this on **Line 281-283**:

The relationship between total soil C stocks and respiratory C losses was inconsistent across these treatments, potentially reflecting differences in the dominant source of the C that was respired ('background' organic matter vs. fresh C inputs via root exudation or surface amendment).

There is an implicit assumption across the literature that soil carbon stabilization is equivalent to net carbon sequestration. This is not the case, as our data clearly show. Not only did we find a positive relationship between mineral-associated C

formation and respiration, but we also demonstrate a *lack* of correlation between mineral C pool sizes and total C pool sizes.

Minor comments:

Title: The title of MS, "Positive correlations between soil carbon stabilization and loss are regulated by roots, minerals and microbes" is not very much informative. The fact that roots, minerals and microbes regulate soil C stabilization is widely accepted and is possibly why the authors considered it as study factor. The only correlation that is present in the MS is related to MAOM C and the cumulative CO₂. In this context, the use of the plural, in this case, appears to be inappropriate.

We have re-named the paper "Relationships among soil carbon stabilization, carbon loss, and carbon stocks are regulated by roots, minerals, and microbes" to more precisely reflect our findings

Balance: during several passages, the authors refer to C balance. I implied they are referring to the variables Total C pool and Cumulative CO₂ (table S1). This needs to be clearly defined.

We have replaced this term throughout the manuscript with more precise terminology (e.g. 'total amount of SOC').

By the way, in table S1 the values of the Total C pool and cumulative CO₂ are similar. Is this a typo, or is it related to the mass balance approach that you used?

This is related to the mass balance approach – in the first phase, most of the variation in C content of each microcosm was related to how much C was lost to respiration, as C inputs hardly varied.

Labile C inputs: In several passages, the authors refer to labile C inputs, but it is rather unclear if they refer to aboveground (different complexities treatments) or to root exudates. One possible confusion is that root C inputs are also formed by labile C inputs (glucose, for example).

We have replaced this terminology with more appropriate language.

Schematic of the experimental approach: The two-step approach is really interesting and innovative, but it is really complicated to keep track of all the factors that were investigated in each step. Fig. 1 is helpful but needs to be improved. I would suggest combining Fig S9 with Fig. 1. Several aspects of the experimental design were only possible to understand after looking into the supplemental material. For example, in the second step, the experimental units received two types of C input (from the root exudates and also aboveground C inputs) at the same time.

We have extensively revised **Figure 1** and its caption to make the experimental design clearer. In particular, we highlight the number of unique treatment combinations (and the number of replicate microcosms per treatment) in each experimental phase.

Validation with real soil and plants: Despite the enormous importance of this for your study, I would keep the validation of the root exudates methods in the supplement materials and just mention it in the Methods section.

We mention the real soils only briefly in the main paper and have retained this language to indicate to the reader that we did have experimental controls in place to verify that biogeochemical dynamics in the artificial soils were similar.

Figure 2: Is there any specific reason for presenting the Log response ratios instead of the original values? Also, are the differences between the mineral reactivity treatments significantly different from the control? What statistical test was used to determine this? A more detailed description of the statistics would be required. Also, what is the control that is being described here? Do you refer to the treatment that did not receive exudates?

We have clarified the experimental control and the statistics used in the caption of **Figure 2**. We used log response ratios because they visualise the significant interactions between the root exudate treatment and mineralogy, clearly showing in which mineral reactivity treatments roots cause an increase (or decrease) a given soil parameter.

Fig.5: We can observe that in most of the treatments, the cumulative C of C was respired. How did you separate the microbial biomass from the remaining SOM fractions? You measured the heavy fraction after the extraction of microbial biomass?

The density fractionation approach we used would separate C that is tightly bound to mineral surfaces from all non-bound C, including that in the microbial biomass, in dissolved organic matter pools, or in particulate form. When different density fractions of soil are incubated separately without re-inoculation of living microbes, C mineralization from the heavy (mineral-associated) fraction is negligible,² which implies that living biomass is not captured by the heavy fraction.

L. 117: Exchange the term dramatic.

Removed

L. 248: The term strong is vague. Also an R^2 of 0.131 shows that the losses of C are not explaining very much the variation in the MAOM formation.

We removed the term 'strong'

Soil respiration: By reading L. 351., I understood that the C loss produced in each microcosm was not measured continuously. I would consider it essential to have more details of how this cumulative C loss was calculated; did you consider daily CO₂ rate uniform among the sampling dates?

We clarified this on **Line 420-422**:

We also calculated cumulative CO₂ respired in each microcosm using the 'area under curve' function in the R package *flux*, which integrates under the curve of CO₂ flux vs. time following the trapezoid rule

We apologize because the previous version of the manuscript erroneously cited the R package *auc*. Instead we used the 'auc' (area-under-curve) function from R package *flux*, now cited here.

Fig S1: Panel C: Lacks a description of the statistics used. Also, does "biomass" refer to "microbial biomass" or something else?

We clarified the statistics and response variables in the caption.

Fig S5. Lack of description of statistics again.

We have revised the caption.

Table S2 and table S3: are analyses described in the statistics section?

Yes, on **Line 414-417** we write:

Three-way ANOVAs were performed to determine effects of artificial root exudates, soil microbial community composition, and soil mineralogy on response variables in Phase 1. ANOVAs on Phase 2 data additionally incorporated effects of non-root C input and date of destructive microcosm harvest (7 months or 13 months).

References cited in this response

1. Wen, T. *et al.* Root exudate chemistry affects soil carbon mobilization via microbial community reassembly. *Fundam. Res.* (2022) doi:10.1016/j.fmre.2021.12.016.
2. Whalen, J. K., Bottomley, P. J. & Myrold, D. D. Carbon and nitrogen mineralization from light- and heavy-fraction additions to soil. *Soil Biol. Biochem.* **32**, 1345–1352 (2000).

Reviewer 3

Major comments:

- In this study, you only evaluated low-mass, soluble C inputs. Other sources of C inputs, such as plant litter, can also form MAOM and may be affected differently. Thus, the validity of the conclusions should be attained for this type of C input. I also consider being necessary to frame this already in the introduction.
- In Fig. 5, there is a fraction named "unprotected C," but it is not clear what this fraction means exactly. As far as I understood, this was not directly measured, and it is the difference between the soil before and after the fractionation. This fraction is normally referred to as non-recovered C and is treated as the quality index of the fractionation procedure. This needs to be correctly addressed in the MS, and I also suggest presenting clearly the C recovery of density fractionation.
- Your graphs show differences between the treatments from the statistical tests even when your means are similar or there is a substantial overlap in the SD of the different treatments. This can be resultant of the high number of observations, which causes an effect size that decreases the p-values. Please, always report the number of observations in each of the graphs.
- In several parts, you affirm that roots, minerals, and microbes interact (L. 132). Nevertheless, your ANOVA tables report several significant interactions that are simply not discussed. I consider that stating that these factors interact is rather superficial and necessary to disentangling how these interactions affect the C dynamics. For example, the harvest (H) factor is completely absent in the discussion. If there is any reason for not going further into the interactions of the factors, this should be explained.

MINOR COMMENTS:

Title: The title, "Relationships among soil carbon stabilization, carbon loss, and carbon stocks are regulated by roots, minerals, and microbes," is still problematic. Saying that roots, plants, and microbes regulate C dynamics is just common sense. It does not encapsulate the manuscript's findings or look interesting from the reader's point of view. This needs to be improved.

ABSTRACT:

L. 17: "which consists mainly of microbial residues bonded to mineral surfaces" Replace "bonded" with "associated"

L. 22: It is not roots resulting in these effects, but the exudates. Please correct this in the rest of MS also.

INTRODUCTION:

L. 31: Longest-lived C seems strange to refer to something mostly dead. Please, exchange the term to long residence time or something analogous.

L. 50: "Key plank." This term looks strange for a scientific manuscript. Please, exchange to a more usual term.

L. 66-67: "Highly organic soils" and "more mineral soils" are unusual terms to describe soils with contrasting C contents. Please rephrase using an adequate description, such as soils with high/low organic C content or even C-rich soils.

L. 76: Belowground C stocks: Replace for soil C stocks to avoid confusion with aboveground/belowground C inputs.

L. 89: "Background" is an unusual term for "native"/Pre-existing SOM. Please, consider replacing it.

L. 121. Please, do not include in the hypothesis measurements that were not evaluated in the experiment, such as Priming effect. This is misleading to the reader.

MATERIAL AND METHODS.

L 418. Statistically significant outliers were removed". Please, provide a more detailed description of how the outliers were handled.

L. 353. It seems that one word is missing in this sentence.

L. 355. I would consider it easier to group all the benchmarking you did in a small separate paragraph. All these benchmarking results are also included in the statistical analysis (correlation etc.)?

L. 365. Any specific reason for increasing this exudation rate?

L. 396. Please, report the recovery of the SOM fractionation in terms of C.
L. 152, 279, Fig. 5 - What do you specifically mean by the mass balance approach? This needs to be described in the MM section.

RESULTS AND DISCUSSION

L. 150: There is no variable named "soil C dynamics" in table S1. Be precise and describe exactly the variables that were (not) affected.

L. 151: Please explain what this total C pool means precisely. This is not clear.

L. 156: Same problem as before with the term "soil C cycling".

L. 152, 153 It is not possible to see this data in this table, which only presents F and p values for the ANOVA. Please, consider providing these results in a more transparent way for the readers.

L. 161. The same problem was reported above. I would consider it essential for you to have a small table with these results, at least in the SI. Only providing the ANOVA tables is not enough.

L. 175. Replace the term recalcitrance (maybe chemical quality would be more appropriate).

L. 176. You do not need to state the number of the experimental units that received a given treatment. Just declare the treatments that are being tested in this phase, and the number of experimental units is implicit.

L. 185 This high proportion of microbial biomass is really concerning. Normally results are 1-5% of soil C. How large is the soil C in these artificial soils? It is difficult to see something in Figure 5 or Figure S6 because the way you report the values is inadequate. It seems that your total soil C is rather small as well. How about comparing the absolute values of microbial biomass with other studies?

L. 190 Change mineral-associated C pools to MAOC in the text. You need to be consistent with the abbreviation used when presenting the results. Also, be careful to use the term pools to refer to fractions. They are not necessarily related and cannot be used interchangeably (<https://doi.org/10.1016/j.soilbio.2007.03.007>).

L. 201 replaces "chemistry" with chemical quality or biochemical quality.

L. 207 replaces "chemistry" with chemical quality or biochemical quality.

L. 218 The interaction between C x R were significant for MAOC, Bacterial and fungal richness. This sentence is not correct. Please, specify what exactly was not significant.

L. 223. Root exudates instead of roots.

L. 224: This sentence gives the idea that root exudates promoted the loss of the MAOC that was previously present in the soil. Is this correct? Or root exudates just resulted in less MAOC at the end of the experiment?

L. 234. Root exudates instead of roots.

L. 236. The root exudates you used are composed of glucose, fructose, and sucrose, which are considered easily available C inputs. They should resemble the chemical composition of the glucose treatment (aboveground). Nevertheless, your results suggest that C cycling disproportionately affected the glucose applied to the surface, even when delivered at similar rates. How can you explain such differences?

L. 247: Kaolinite is not uncharged, I am afraid that is incorrect. It has pH-dependent surface charges.

L. 258: This is an interesting point.

L. 261. I still consider that the hypothesis about the correlation (H3) is not very interesting/novel. It is just confirming the MEMS framework. If microbial byproducts are the source for MAOC formation, then the increase in microbial activity (CO₂ fluxes) would also lead to MAOC formation. Also, your correlation has a really low R², suggesting several other factors also affect this process. Disentangling the factors that affect this process should be much more interesting. For example, is this correlation valid for all soil types and C inputs that you tested? This should be explicitly discussed in the text.

L. 261. Yes, they are expected to interact, and this is just common sense. You need to dig deeper into your data to discuss how these factors affect each other.

L. 280. It is not possible to observe this in Table S5. You need to present the means of the results and not only the ANOVA table.

L. 283: Change the term "background", because it is not usual.

L. 289-292: In this experiment there was an input of large amounts of fresh C up to the end of the experiment, right? Maybe there was not enough time to decompose this fresh C, which remained in the microbial biomass and unprotected fractions. So, in this case, I consider that this effect is more related to experimental conditions and has a limited implication to explain soil C stocks in a generic way.

L. 298: Change the term "belowground".

Figure 1. I still think that this figure can be improved.

Figure 2. From the information in this graph, it is impossible to say whether the differences between the control and the treatment with root are statistically significant. It is important to specify when they are significant, and what post-hoc test/probability level you used.

Figure 4. Please consider changing the units of the axis for mg g of soil. A big number of digits after the comma is not very usual and hampers the interpretation of the results.

Figure 5. Please, increase the size of the numbers on the y-axis. Also, in the y-axis, the caption "g C in each pool" is g of C per g of soil, right?

Table S1: Microbial biomass is reported as microgram of C, right?

Figure S1: Consider using MAOC in the caption of the y-axis, as in other text parts.

Figure S6: It is more common to report MAOC in mg C g⁻¹ of soil. I would consider it interesting to change this to facilitate the visualization of the graphs. The same can be said for microbial biomass C and CO₂-C.

CONCLUSIONS

209: In this study, the destabilization of MAOC was not measured. You only reported the total flux of CO₂, which can also come from the decomposition of the unprotected C that you added to the soil. I do not consider that this study has evidence that this process is happening, so I would keep this out.

301: "But independent of total C stocks": I do not see how your correlation proves this. Please clarify.

309: The mechanisms that control the preservation/accumulation of "unprotected" C in these specific soils (tundra, peatlands) are related to restrictions in microbial activity given by temperature/hypoxia and have nothing to do with the system you tested. In your experiment, you have a situation with high abundance of C soluble inputs into the soil. In nature, I would assume that the preservation of C in soluble/unprotected forms would be really small, considering that soil microbes are mostly C-restricted. Please re-adequate the conclusions to apply for situations comparable to the tested conditions.

We thank the Editor and Reviewer for their insightful comments and for the opportunity to revise our manuscript. Please see our point-by-point responses below

Reviewer 3

Major comments:

- In this study, you only evaluated low-mass, soluble C inputs. Other sources of C inputs, such as plant litter, can also form MAOM and may be affected differently. Thus, the validity of the conclusions should be attained for this type of C input. I also consider being necessary to frame this already in the introduction.

Thank you for this suggestion – we have amended the manuscript in a few places. For example, on **Line 133** in the introduction, we specifically indicate our experiment examined individual C compounds (not whole litters) and on **Line 215-217** in the Discussion we write:

It is also possible that amending soils with individual compounds (rather than whole plant litters) obscured ecological dynamics observed in real ecosystems, e.g. microbial community specialization on particular plant types.

However, we note that our carbon amendments were specifically designed to capture a gradient of C input chemical complexity. Xylan, the most ‘recalcitrant’ input we used, is neither soluble at room temperature, nor low in molecular weight, with molecular mass > 20 kDa. (Note: to apply the xylan to the microcosms, we kept the solution under continuous agitation to ensure a homogenous quantity of substrate was delivered in each aliquot, even though it did not completely dissolve.)

- In Fig. 5, there is a fraction named "unprotected C," but it is not clear what this fraction means exactly. As far as I understood, this was not directly measured, and it is the difference between the soil before and after the fractionation. This fraction is normally referred to as non-recovered C and is treated as the quality index of the fractionation procedure. This needs to be correctly addressed in the MS, and I also suggest presenting clearly the C recovery of density fractionation.

We apologize that the previous version of our manuscript was unclear on this point. Unprotected C is **not** the same as non-recovered C in our fractionations. In fact, recovery in our fractionation approached 100% (**Line 402-406**):

On average, we recovered 100.4 ± 0.2 % of the mass of soil subject to density fractionation - that is, the mass of the isolated heavy and light fractions summed to the mass of the whole soil sample. This indicates near-perfect preservation of the sample (no C loss) and very minor contamination of the sample with sodium polytungstate.

Rather, the ‘unprotected C’ was calculated as follows (**Line 408-417**):

Because light fraction masses were so low, we did not quantify C content of these samples directly. Rather, the quantity of ‘unprotected C’ in each microcosm (in particulate and dissolved organic matter) was calculated via a mass balance approach. We determined ‘total available C’ in each microcosm as the sum of C contained in the ground corn leaves, root exudates and surface amendments of glucose, cellobiose, or xylan. We then calculated ‘recovered C’ – the sum of cumulative C lost to respiration, microbial biomass C, and heavy fraction C (MAOC) – quantities determined empirically for each microcosm. The difference between total available C and recovered C was termed ‘unprotected C,’ and is assumed to reflect particulate C that was not incorporated into the microbial biomass or stabilized on mineral surfaces at the time of measurement.

- Your graphs show differences between the treatments from the statistical tests even when your means are similar or there is a substantial overlap in the SD of the different treatments. This can be resultant of the high number of observations, which causes an effect size that decreases the p-values. Please, always report the number of observations in each of the graphs.

We have done so for all figures and figure captions.

- In several parts, you affirm that roots, minerals, and microbes interact (L. 132). Nevertheless, your ANOVA tables report several significant interactions that are simply not discussed. I consider that stating that these factors interact is rather superficial and necessary to disentangling how these interactions affect the C dynamics. For example, the harvest (H) factor is completely absent in the discussion. If there is any reason for not going further into the interactions of the factors, this should be explained.

Due to the large number of response variables and interacting predictors in the models, there simply is not space to discuss all significant interactions. We only mention those with **biologically** significant results, which are clearly visible when the raw data are plotted. As an example, below is a visualization of the interactions between harvest (7 vs. 13 months, solid vs. dashed lines) and (on the x axis in each panel, left to right): the root exudate treatment, the inoculum treatment, and the carbon treatment. As you can see, while the interactions are significant, the effect sizes are negligible.

However, there were a few interactions which did have notable effects, and which we had not explicitly highlighted in previous manuscript versions. We have now done so on **Line 192**:

Although respiration rates did not differ between inoculum treatments in the second experimental phase, MAOC pools were larger in microcosms which were inoculated with FB vs. BO (**Fig S6**), especially in the cellulose treatment.

And we now discuss all root-carbon treatment interactions, as follows on **Line 222-226**:

... MAOC pools in microcosms amended with glucose were 25% lower in the presence of root exudates. Diversity and structure of bacterial communities responded to a root-carbon interaction as well (**Table S3, S4**), suggesting this MAOC response was microbially mediated, perhaps reflecting a shift towards a fast-growing copiotrophic community.

MINOR COMMENTS:

Title: The title, "Relationships among soil carbon stabilization, carbon loss, and carbon stocks are regulated by roots, minerals, and microbes," is still problematic. Saying that roots, plants, and microbes regulate C dynamics is just common sense. It does not encapsulate the manuscript's findings or look interesting from the reader's point of view. This needs to be improved.

We revised the title to: "Mineral reactivity determines root effects on the stabilization and mineralization of soil organic carbon"

ABSTRACT:

L. 17: "which consists mainly of microbial residues bonded to mineral surfaces"
Replace "bonded" with "associated"

Done

L. 22: It is not roots resulting in these effects, but the exudates. Please correct this in the rest of MS also.

Good point; done

INTRODUCTION:

L. 31: Longest-lived C seems strange to refer to something mostly dead. Please, exchange the term to long residence time or something analogous.

Done

L. 50: "Key plank." This term looks strange for a scientific manuscript. Please, exchange to a more usual term.

Done

L. 66-67: "Highly organic soils" and "more mineral soils" are unusual terms to describe soils with contrasting C contents. Please rephrase using an adequate description, such as soils with high/low organic C content or even C-rich soils.

Done

L. 76: Belowground C stocks: Replace for soil C stocks to avoid confusion with aboveground/belowground C inputs.

Done

L. 89: "Background" is an unusual term for "native"/Pre-existing SOM. Please, consider replacing it.

Done

L. 121. Please, do not include in the hypothesis measurements that were not evaluated in the experiment, such as Priming effect. This is misleading to the reader.

We deleted the reference to the priming effect

MATERIAL AND METHODS.

L 418. Statistically significant outliers were removed". Please, provide a more detailed description of how the outliers were handled.

We have done so on **Line 440**

L. 353. It seems that one word is missing in this sentence.

The sentence is edited

L. 355. I would consider it easier to group all the benchmarking you did in a small separate paragraph. All these benchmarking results are also included in the statistical analysis (correlation etc.)?

We analysed data from the 'real soils' and artificial soils separately, and the results for the real soils are shown in **Fig S1**

L. 365. Any specific reason for increasing this exudation rate?

In the second experimental phase, we aimed to add comparable amounts of C via surface amendments and through root exudates

L. 396. Please, report the recovery of the SOM fractionation in terms of C.

We have done so on **Line 402** (recovery averaged 100%)

L. 152, 279, Fig. 5 - What do you specifically mean by the mass balance approach? This needs to be described in the MM section.

Done, as described above

RESULTS AND DISCUSSION

L. 150: There is no variable named "soil C dynamics" in table S1. Be precise and describe exactly the variables that were (not) affected.

We replaced this term with 'respiration' on **Line 156**

L. 151: Please explain what this total C pool means precisely. This is not clear.

We have clarified this in the methods (**Line 417-420**):

Furthermore, we refer to the difference between 'total available C' and cumulative respiration as the 'total C pool' in each microcosm – this invokes the assumption that

no C losses occurred other than through respiration (which is appropriate as the microcosms were closed systems).

L. 156: Same problem as before with the term "soil C cycling".

Fixed

L. 152, 153 It is not possible to see this data in this table, which only presents F and p values for the ANOVA. Please, consider providing these results in a more transparent way for the readers.

This pattern can be seen in **Fig S3**, to which we now refer the reader

L. 161. The same problem was reported above. I would consider it essential for you to have a small table with these results, at least in the SI. Only providing the ANOVA tables is not enough.

We added **Table S1b** to provide the CUE data the Reviewer is referring to here.

L. 175. Replace the term recalcitrance (maybe chemical quality would be more appropriate).

Done

L. 176. You do not need to state the number of the experimental units that received a given treatment. Just declare the treatments that are being tested in this phase, and the number of experimental units is implicit.

L. 185 This high proportion of microbial biomass is really concerning. Normally results are 1-5% of soil C. How large is the soil C in these artificial soils? It is difficult to see something in Figure 5 or Figure S6 because the way you report the values is inadequate. It seems that your total soil C is rather small as well. How about comparing the absolute values of microbial biomass with other studies?

We have specifically acknowledged (**Line 185-188**) that microbial biomass occupies a large portion of the total C pool (which was relatively modest - 0.017 kg C kg⁻¹ soil), and provide some reasons why this might be so. The absolute value of microbial biomass (shown in **Fig S2**) averaged about 3000 µg C g⁻¹ soil. This falls well within the range of observed microbial biomass C values in a meta-analysis of global-scale patterns in microbial abundance (<https://doi.org/10.1016/j.catena.2022.106037>).

L. 190 Change mineral-associated C pools to MAOC in the text. You need to be consistent with the abbreviation used when presenting the results. Also, be careful to use the term pools to refer to fractions. They are not necessarily related and cannot be used interchangeably (<https://doi.org/10.1016/j.soilbio.2007.03.007>).

We have used the term 'MAOC' throughout. We feel that our use of 'pool' is correct because, since we know the total mass of soil in each microcosm and the fraction of that mass that is in the form of MAOC, we can report the total mass (in g) of MAOC – that is, the pool size. This is clarified in the legend of **Fig 5**.

L. 201 replaces "chemistry" with chemical quality or biochemical quality.
L. 207 replaces "chemistry" with chemical quality or biochemical quality.

Done, in both places

L. 218 The interaction between C x R were significant for MAOC, Bacterial and fungal richness. This sentence is not correct. Please, specify what exactly was not significant.

We have elaborated on these interactions (**Line 222-226**, and quoted above)

L 223. Root exudates instead of roots.

Fixed

L. 224: This sentence gives the idea that root exudates promoted the loss of the MAOC that was previously present in the soil. Is this correct? Or root exudates just resulted in less MAOC at the end of the experiment?

The latter; we have revised the sentence accordingly

L 234. Root exudates instead of roots.

Fixed

L 236. The root exudates you used are composed of glucose, fructose, and sucrose, which are considered easily available C inputs. They should resemble the chemical composition of the glucose treatment (aboveground). Nevertheless, your results suggest that C cycling disproportionately affected the glucose applied to the surface, even when delivered at similar rates. How can you explain such differences?

The two treatments are not directly comparable, because the delivery of exudate C was highly localized to the zone immediately around the root. Also, the exudates contained other elements, including nitrogen (in the tryptic soy broth, the fourth component of the exudate solution, **Line 367**.) We considered contrasting the effects of root exudate vs. 'labile' C additions (glucose/cellobiose) more extensively in the discussion. However, our experimental design does not permit this direct comparison, because during the second experimental phase, we did not have microcosms which received only root exudates or only surface glucose amendments. Root exudates always occurred in the presence of one other type of carbon amendment.

L. 247: Kaolinite is not uncharged, I am afraid that is incorrect. It has pH-dependent surface charges.

This is a good point – technically all clays exhibit variable charge depending upon soil pH, but in the typical soil pH range (4-8), kaolinite and montmorillonite tend to be negatively charged and goethite positively charged (although the PZC for goethite can occur within this range). We have therefore re-written the sentence as follows (**Line 253-256**):

However, both phyllosilicate minerals differ from goethite in a critical respect: in contrast with 1:1 phyllosilicate minerals like kaolinite and 2:1 phyllosilicate minerals like montmorillonite, which are negatively charged under normal soil pH conditions, metal oxides like goethite exhibit variable charge and bear highly reactive hydroxyl groups.

L. 258: This is an interesting point.

L. 261. I still consider that the hypothesis about the correlation (H3) is not very interesting/novel. It is just confirming the MEMS framework. If microbial byproducts are the source for MAOC formation, then the increase in microbial activity (CO₂ fluxes) would also lead to MAOM formation. Also, your correlation has a really low R², suggesting several other factors also affect this process. Disentangling the factors that affect this process should be much more interesting. For example, is this correlation valid for all soil types and C inputs that you tested? This should be explicitly discussed in the text.

We did not discuss the relationship with reference to the other treatments because the results were the same when data were analysed independently for kaolinite soils ($\beta = 0.140$, $R^2 = 0.093$, $P = 0.004$), montmorillonite soils ($\beta = 0.143$, $R^2 = 0.128$, $P < 0.001$), and goethite soils ($\beta = 0.090$, $R^2 = 0.229$, $P = 0.011$). The patterns were also the same for soils amended with glucose ($\beta = 0.187$, $R^2 = 0.254$, $P < 0.001$), cellobiose ($\beta = 0.140$, $R^2 = 0.112$, $P = 0.006$), and xylan ($\beta = 0.159$, $R^2 = 0.122$, $P = 0.012$). To help the reader visualise this, in **Fig 4**, we have now indicated soil mineral type with symbols.

L. 261. Yes, they are expected to interact, and this is just common sense. You need to dig deeper into your data to discuss how these factors affect each other.

This is discussed extensively on **Line 243-267**, and **285-299**

L. 280. Its is not possible to observe this in Table S5. You need to present the means of the results and not only the ANOVA table.

We point the reader to **Fig 5** where this data can be examined directly

L. 283: Change the term "background", because it is not usual.

Done

L. 289-292: In this experiment there was an input of large amounts of fresh C up to the end of the experiment, right? Maybe there was not enough time to decompose this fresh C, which remained in the microbial biomass and unprotected fractions. So, in this case, I consider that this effect is more related to experimental conditions and has a limited implication to explain soil C stocks in a generic way.

In reality, soils are constantly receiving fresh C inputs – via litterfall, root exudation, root mortality, etc. – even as older inputs are still undergoing decomposition. Our experimental design is therefore not unusual or unrepresentative of natural C cycling processes. Our point is that the size of the soil carbon pool is determined by the

balance between inputs and outputs. Therefore, increases or decreases in the rate of soil respiration do not necessarily reveal whether the entire soil C pool is increasing or decreasing. A similar insight emerges from the network of DIRT (Detrital Input and Removal Treatment) experiments, and other real-world studies. However, our experiment provides a unique contribution by allowing us to explore, with a high degree of mechanistic detail, how this decoupling is mediated by soil mineralogy, microbes, root exudates, and the chemistry of C inputs.

L. 298: Change the term "belowground".

Done

Figure 1. I still think that this figure can be improved.

We have modified the legend to help the reader interpret the figure

Figure 2. From the information in this graph, it is impossible to say whether the differences between the control and the treatment with root are statistically significant. It is important to specify when they are significant, and what post-hoc test/probability level you used.

This figure is visualising the interaction between root exudate and mineral treatments, so we have provided the appropriate statistical results (F and P values) directly on the figure.

Figure 4. Please consider changing the units of the axis for mg g of soil. A big number of digits after the comma is not very usual and hampers the interpretation of the results.

Done

Figure 5. Please, increase the size of the numbers on the y-axis. Also, in the y-axis, the caption "g C in each pool" is g of C per g of soil, right?

Done. The figure refers to the total amount of C, in g, residing in a particular pool (e.g. MAOC) in each microcosm. It is not g per g soil.

Table S1: Microbial biomass is reported as microgram of C, right?

It is reported as microgram C per gram soil, as shown in the table

CONCLUSIONS

209: In this study, the destabilization of MAOC was not measured. You only reported the total flux of CO₂, which can also come from the decomposition of the unprotected C that you added to the soil. I do not consider that this study has evidence that this process is happening, so I would keep this out.

We have removed reference to 'destabilization'

301: "But independent of total C stocks": I do not see how your correlation proves this. Please clarify.

We revised the sentence for clarity

309: The mechanisms that control the preservation/accumulation of "unprotected" C in these specific soils (tundra, peatlands) are related to restrictions in microbial activity given by temperature/hypoxia and have nothing to do with the system you tested. In your experiment, you have a situation with high abundance of C soluble inputs into the soil. In nature, I would assume that the preservation of C in soluble/unprotected forms would be really small, considering that soil microbes are mostly C-restricted. Please re-adequate the conclusions to apply for situations comparable to the tested conditions.

We agree and so have removed references to tundra/peatlands